# Short-term outcomes of cochlear implantation for single-sided deafness compared to bone conduction devices and contralateral routing of sound hearing aids— Results of a Randomised controlled trial (*CINGLE-trial*)

Jeroen P. M. Peters[1,2], Jan A. A. van Heteren[1,2ʘ], Anne W. Wendrich[1,2ʘ], Gijsbert A. van Zanten[1,2], Wilko Grolman[3], Robert J. Stokroos[1,2], Adriana L. Smit[1,2]*

**1** Department of Otorhinolaryngology and Head & Neck Surgery, University Medical Center Utrecht, Utrecht, The Netherlands, **2** UMC Utrecht Brain Center, University Medical Center Utrecht, Utrecht, The Netherlands, **3** Causse Ear Clinic, Colombiers, France

ʘ These authors contributed equally to this work.
* a.l.smit-9@umcutrecht.nl

## Abstract

Single-sided deafness (SSD) leads to difficulties with speech perception in noise, sound localisation, and sometimes tinnitus. Current treatments (Contralateral Routing of Sound hearing aids (CROS) and Bone Conduction Devices (BCD)) do not sufficiently overcome these problems. Cochlear implants (CIs) may help. Our aim was to evaluate these treatments in a Randomised Controlled Trial (RCT). Adult SSD patients were randomised using a web-based randomisation tool into one of three groups: CI; trial period of 'first BCD, then CROS'; trial period of 'first CROS, then BCD'. After these trial periods, patients opted for BCD, CROS, or No treatment. The primary outcome was speech perception in noise (directed from the front ($S_0N_0$)). Secondary outcomes were speech perception in noise with speech directed to the poor ear and noise to the better ear ($S_{pe}N_{be}$) and vice versa ($S_{be}N_{pe}$), sound localisation, tinnitus burden, and disease-specific quality of life (QoL). We described results at baseline (unaided situation) and 3 and 6 months after device activation. 120 patients were randomised. Seven patients did not receive the allocated intervention. The number of patients per group after allocation was: CI (n = 28), BCD (n = 25), CROS (n = 34), and No treatment (n = 26). In $S_0N_0$, the CI group performed significantly better when compared to baseline, and when compared to the other groups. In $S_{pe}N_{be}$, there was an advantage for all treatment groups compared to baseline. However, in $S_{be}N_{pe}$, BCD and CROS groups performed worse compared to baseline, whereas the CI group improved. Only in the CI group sound localisation improved and tinnitus burden decreased. In general, all treatment groups improved on disease-specific QoL compared to baseline. This RCT demonstrates that cochlear implantation for SSD leads to improved speech perception in noise,

**Data Availability Statement:** All relevant data are within the paper and its Supporting Information files.

**Funding:** This study is partly funded by Cochlear Ltd. as an unrestricted research grant. By research contract, Cochlear Ltd. did not have influence on the study design, data collection, analysis, data interpretation, and publication. The funders had no role in study design, data collection and analysis, decision to publish, or preparation of the manuscript.

**Competing interests:** There are no competing interests for any of the authors. This study is partly funded by Cochlear Ltd. as an unrestricted research grant. By research contract, Cochlear Ltd. did not have influence on the study design, data collection, analysis, data interpretation, and publication. This does not alter our adherence to PLOS ONE policies on sharing data and materials.

sound localisation, tinnitus burden, and QoL after 3 and 6 months of follow-up. For most outcome measures, CI outperformed BCD and CROS.

**Trial registration:** Netherlands Trial Register (www.trialregister.nl): NTR4580, *CINGLE-*trial.

## Introduction

### Single-sided deafness

Single-sided deafness (SSD) is defined as severe to profound hearing loss in one ear and near to near-normal hearing in the contralateral ear [1]. Patients with SSD often experience problems with speech perception in noise and sound localisation, because they lack binaural benefits [2]. Specifically, they cannot benefit from the binaural summation effect (redundancy of auditory input) and the squelch effect (the ability of the brain to suppress noise on the side with the best signal-to-noise-ratio using the noise information of the contralateral side) [2,3]. Moreover, the head acts as an acoustic barrier and attenuates signals from the contralateral side (head shadow effect) [2,4]. The head shadow effect can be of benefit for SSD patients when the speaker is on the side of the better hearing ear, but it may deteriorate speech perception when the speaker is on the deaf side. Furthermore, SSD patients frequently suffer from tinnitus in the affected ear [2,5]. Due to these hearing-related difficulties, patients with SSD may experience problems in social interaction and communication, possibly negatively affecting their quality of life (QoL) [6–8].

### CROS and BCDs

In most countries, there are currently two reimbursed treatment options for patients with SSD: Contralateral Routing of Sound hearing aids (CROS) and Bone Conduction Devices (BCDs). A CROS transfers signals from the hearing field of the poor ear to an output transducer in the ear canal of the better ear, so that awareness for sound at the poor side is restored. A BCD conducts signals from the hearing field of the poor side to the cochlea of the better ear by vibration of the skull bone via a titanium implant or tight headband. CROS and BCD can alleviate the head shadow effect and restore sound awareness from the impaired side [9,10]. However, neither modality provides auditory input to the deaf ear, and thus binaural hearing cannot be restored [10].

So far, only retrospective case series comparing CROS and BCD for SSD have been published (for a review see Peters et al. [11]). The included studies often had small sample sizes, unclear inclusion criteria and a moderate to high risk of bias. As demonstrated in this review [11], CROS and BCD only led to improved speech perception in noise when speech was presented to the impaired ear. Both treatment modalities and the unaided condition show equal results for sound localization. Quality of life did not differ significantly between conditions, but quality of hearing improved for the speech communication subscales of the Abbreviated Profile for Hearing Aid Benefit (APHAB) questionnaire.

### Cochlear implantation

Since a cochlear implant (CI) provides input to the auditory nerve of the deaf ear, binaural input can be partially restored. Consequently, cochlear implantation led to improved speech perception in noise and sound localisation in patients with SSD [12–14].

Tinnitus is frequently believed to be the consequence of (central) auditory deprivation due to hearing loss. Cochlear electrical stimulation of the auditory nerve is assumed to have a beneficial influence on tinnitus burden [15]. Recent studies reported lower tinnitus burden following cochlear implantation in SSD patients [16–18].Moreover, (disease-specific) quality of life of SSD patients was also improved following cochlear implantation [13,17,19].

However, previously published studies on CI for SSD often had retrospective study designs, unclear eligibility criteria and definitions of SSD, small sample sizes with lack of statistical power, or no power calculations (see reviews [9,17,20]). Although recently published studies showed improved methodology with larger sample sizes and prospective trials design, randomised allocation of treatment to ensure equal distribution of known and unknown confounders amongst groups was not performed.

### Need for current trial

To provide higher level of evidence about the treatment outcomes of CI, BCD, and CROS for SSD, we designed a Randomised Controlled Trial (RCT). In this paper we publish the short-term results of our RCT comparing CI, BCD, CROS, and No treatment for patients with SSD. The outcomes of interest were speech perception in noise, sound localisation, tinnitus burden, and disease-specific QoL at 3 and 6 months follow-up (out of total follow-up duration of 5 years). Later follow-up moments as well as other secondary outcomes (e.g. generic QoL, cost utility) will be discussed later.

## Materials and methods

The research protocol of this study was approved by the Institutional Review Board of the University Medical Center Utrecht (NL45288.041.13) and is registered in the Netherlands Trial Register (www.trialregister.nl, NTR4580). For a detailed description of the *CINGLE*-trial (*Cochlear Implantation for siNGLE-sided deafness*), we refer to the study protocol [21] or **S1 File**. The authors confirm that all ongoing and related trials for this drug/intervention are registered. Details about patient recruitment, sample size calculation, and ethical considerations are described here. All patients provided signed Informed Consent between July 2014 and February 2019, and the last follow-up visit took place in March 2020. After approval of the study protocol, but before the first patient was included in the trial, we amended the trial protocol to also measure at 3 months follow-up. In the current paper we presented a summary of the methods to assess the presented outcomes. This RCT is reported according to the CONSORT guidelines [22], and a separate CONSORT Checklist is uploaded as **S2 File**.

### Study population

Patients were eligible for inclusion if they fulfilled the following criteria:

- age 18 years or older;

- Pure Tone Average threshold at 0.5, 1, 2, 4 kHz: of the best ear ($PTA_{be}$) maximum 30 dB HL, and of the poor ear minimum 70 dB HL;

- duration of SSD minimum 3 months and maximum 10 years;

- health status allows general anaesthesia and surgery for the potential implantation of BCD or CI

- Dutch language proficiency

- coverage of Dutch health insurance

• willingness and ability to participate in all scheduled procedures outlined in the protocol

Patients were not eligible for inclusion if they had retrocochlear pathology or abnormal cochlear anatomy (e.g. ossification) or an implanted BCD. Patients were referred to our tertiary clinic by otolaryngologists or audiologists who were informed of our trial at national meetings. Patients were also informed of our trial via announcements on the websites of SSD patient associations, and could contact the research team directly.

## Study design

Patients were randomised in one of three groups by a web-based randomisation tool (ratio 2:3:3, block randomisation):

• CI (type: Cochlear™);

• trial period of first BCD on headband, then CROS;

• trial period of first CROS, then BCD on headband (see also **Fig 1**).

In the latter two groups (BCD-CROS, CROS-BCD), each device was tested for six weeks. The reversed order of the trial periods in the BCD-CROS and CROS-BCD groups was implemented to correct for the order effect: patients may judge their second hearing aid based on experiences with the first hearing aid. After the trial period, patients were asked which of both treatments they preferred: when they opted for BCD, the abutment was surgically implanted

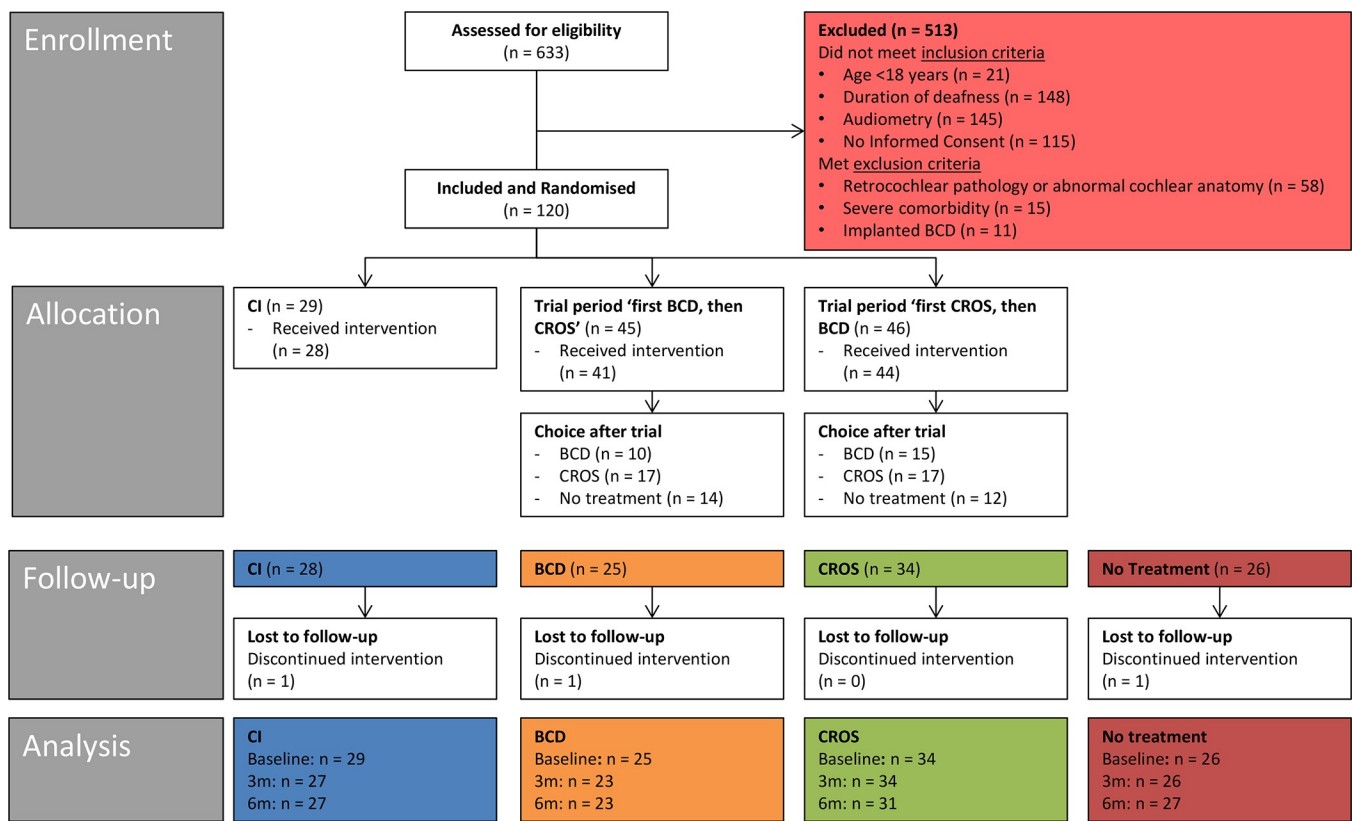

**Fig 1. CONSORT flow diagram of the study.** See S3 File for an explanation of the numbers per group at different moments.

and after six weeks fitted with a BCD (Cochlear™). Patients who opted for a CROS were referred to a local audiologist to purchase a CROS, and later device settings were optimized by an experienced audiologist from our audiological center. Patients could also opt for 'No treatment' if neither BCD nor CROS was preferred. This trial period and the possible options resemble the current clinical practice in The Netherlands for single-sided deaf patients. Following the trial period, four groups were assessed:

- CI;

- BCD;

- CROS;

- No treatment.

In this paper, we report on data obtained at baseline (i.e. unaided situation), at 3, and at 6 months follow-up (i.e. after activation of CI, fitting of CROS or BCD, or unaided after the end of the trial period for patients that opted for no treatment, respectively).

## Outcomes

Speech perception in noise and sound localisation were measured in the Crescent of Sound test set-up [23] following the same protocol. Patients were instructed not to move their head.

**Speech perception in noise.** The primary outcome was speech perception in noise, with speech and noise coming front the front ($S_0N_0$) with the *Utrecht Sentence Test with Adaptive Randomized Roving Levels* (U-STARR) [24]. The U-STARR is designed to determine a patient's ability to understand speech in a noisy environment. Five key words were selected in long sentences and 3 key words in shorter sentences. A sentence was considered to be understood correctly when a subject repeated at least 3 out of 5 or 2 out of 3 key words correctly. Sentences were presented at 65, 70, or 75 dB SPL (randomly selected) with an initial signal-to-noise-ratio (SNR) of +20 dB (sentence 20 dB louder than noise). When the sentence was repeated correctly, the SNR of the next sentence was decreased. If repeated incorrectly the SNR for the next sentence was increased. The SNR step size was adaptively decreasing from initially 10 to 5, and to 2.5 dB. At the latter size at least 10 sentences were presented, the mean of which was the measurement outcome. Noise was presented 500 ms before the start of the sentence and ended 500 ms after the sentence. Additionally, speech perception in noise was measured with speech directed to the poor ear (pe) and noise directed to the better ear (be) and with speech directed to the better ear (be) and noise directed to the poor ear (pe) ($S_{pe}N_{be}$ and $S_{be}N_{pe}$, respectively), with the stimuli coming from -60˚ and +60˚. The outcome was a speech reception threshold in noise (SRTn) in dB at which patients were able to repeat 50% of the sentences correctly [24].

**Sound localisation.** The stimulus (sentence "hello, what's this?", spoken by various female English speakers) for the sound localisation task was presented in quiet (roving level of 55–65 dB SPL) in three configurations: 5 loudspeakers separated by an angle of 15˚ (-30˚, -15˚, 0˚, +15˚, and +30˚); 5 loudspeakers separated by an angle of 30˚ (-60˚, -30˚, 0˚, +30˚, and +60˚); 3 loudspeakers separated by an angle of 60˚ (-60˚, 0˚, and +60˚). The patient had to indicate from which loudspeaker the stimulus came. The outcome was a percentage correct score (30 sentences per configuration were presented).

**Tinnitus.** At baseline and at 3 and 6 months follow-up, patients indicated whether they had tinnitus or not. If yes, tinnitus questionnaires were completed. In this paper we report outcomes of the validated Dutch versions of the Tinnitus Questionnaire (TQ) as a self-report measure of tinnitus-related distress [25,26] and of the Tinnitus Handicap Inventory (THI) that quantifies the impact of tinnitus on daily life [27].

**Disease-specific QoL.** Patients completed three disease-specific QoL questionnaires:

- The Speech, Spatial and Qualities of hearing (SSQ) scale [28] assesses three domains of hearing in everyday life.

- The APHAB questionnaire [29] has four subdomains: ease of communication (EC), listening under reverberant conditions (RV), listening in background noise (BN), and aversiveness of sounds (AS).

- The Glasgow Benefit Inventory (GBI) is validated to measure outcomes on health status after otorhinolaryngological procedures [30]. The GBI is scored on three subscales (general, social support, and physical health).

## Statistical analysis

None of the outcome variables were normally distributed, so we used non-parametric tests. We used a generalized estimating equation (mixed model) to compare values within groups over time and between groups at follow-up moments. Due to the nature of our interventions, subject data were analysed 'as treated'. All figures show boxplots with medians, $1^{st}$ and $3^{rd}$ quartiles, and minimum/maximum values to show the distribution of data.

Because there was a low percentage of missing data for all outcomes, we chose not to use multiple imputation as was stated in the study protocol. Instead, we performed a complete case analysis (missing data: speech perception in noise 0.9%, sound localisation 0.9%, TQ 3.5%, THI 3.5%, SSQ 1.2%, APHAB 3.8%, and GBI 1.9%). A statistically significant result was defined as a *p*-value <0.05. Statistical package SPSS version 25.0 (IBM Corp., Armonk, NY, USA) was used.

## Results

### Patient characteristics

We assessed 633 patients for eligibility. Of these, 513 patients were excluded due to various reasons (see Flow diagram in **Fig 1**). In total, 120 patients were randomised into three groups: 29 patients to the CI group, 45 patients to the 'first BCD, then CROS' group, and 46 patients to the 'first CROS, then BCD' group. Not all patients received the allocated intervention; for an explanation of the numbers per group, see **S3 File**.

**Table 1** shows the patient characteristics of our study population per group after randomisation. The mean age at inclusion of the complete study sample was 53.0 ± 12.1 (standard deviation, SD) years, the median $PTA_{be}$ was 15.0 dB (range 2.5–30.0 dB) and the median duration of deafness was 1.8 years (range 3 months– 10 years).

The patient characteristics of the groups after allocation of treatment (i.e. CI activation, BCD implantation and CROS fitting) and for the No treatment group are displayed in **S1 Table**. There were no significant differences in patient characteristics between these groups ($p > 0.05$).

### Device characteristics

Due to the long inclusion period of the study, newer versions of the CI and BCD became available during the study. The first 12 CI patients were implanted with a Nucleus® CI422 with a Slim Straight electrode array, the other 16 patients were implanted with a Nucleus® CI512 with a Contour Advance electrode array. The median duration of CI use per day during the

**Table 1. Patient characteristics per group after randomisation.**

| | CI | BCD, CROS | CROS, BCD | Statistics |
|---|---|---|---|---|
| **Gender** | | | | |
| Male:Female | 14:15 | 24:21 | 20:26 | ns[a] |
| **Age at inclusion (years)** | | | | |
| Mean (SD) | 53.2 (13.1) | 53.6 (12.6) | 52.3 (11.2) | ns[b] |
| **PTA$_{be}$ (0.5–4 kHz) (dB)** | | | | |
| Mean (SD) | 15.6 (7.1) | 15.6 (6.9) | 15.1 (6.8) | ns[c] |
| Median [range] | 15.0 [5.0–30.0] | 15.0 [6.3–30.0] | 15.6 [2.5–28.8] | |
| **Duration of deafness (years)** | | | | |
| Mean (SD) | 3.1 (2.8) | 3.1 (3.3) | 2.7 (2.7) | ns[c] |
| Median [range] | 1.9 [0.3–10.0] | 1.3 [0.3–10.0] | 1.8 [0.3–10.0] | |
| **Etiology** | | | | |
| Unknown | 6 | 10 | 13 | ns[a] |
| Iatrogenic | 1 | 1 | 0 | |
| Sudden deafness | 15 | 21 | 23 | |
| Labyrinthitis | 4 | 4 | 4 | |
| Infection *(not otherwise specified)* | 0 | 3 | 1 | |
| Ménière's disease | 3 | 4 | 3 | |
| Traumatic | 0 | 2 | 2 | |

Abbreviations:

CI = Cochlear Implant, BCD = Bone Conduction Device, CROS = Contralateral Routing of Sound hearing aid, SD = standard deviation, PTA$_{be}$ = Pure Tone Average threshold of the best ear at 0.5, 1, 2, 4 kHz (dB), ns = not significant.

[a] Fisher's Exact test.

[b] One-way ANOVA.

[c] Kruskal Wallis test.

first 6 months after fitting was 12.7 hours (range 2.0–16.3 hours), based on the electronic data logging of the speech processor. There were no non-users in the CI group.

In the trial period, the first 52 patients tested the Baha® BP110 on a headband from 2014 until 2016. From 2017 on, 33 patients tested the Baha® 5 Power on a headband. There was a remarkable difference in satisfaction after the trial period with these two devices: 10 of 52 patients (19.2%) chose to proceed with BCD after the Baha® BP110 trial, whereas 15 of 33 patients (45.5%) opted for BCD after the Baha® 5 Power trial (Fisher's exact test, *p* = 0.014). Thus, in total 25 of 85 patients (29.4%) opted to be implanted with a BCD after the trial period. After implantation, three patients were fitted with the Baha® BP110 and all other patients were fitted with the Baha® 5 Power.

## Speech perception in noise

The results on speech perception in noise are shown in **Figs 2–4**. Lower values (SRTn in dB) reflect better performance. See **S2 Table** for exact data.

**$S_0N_0$.** For the CI group, there was a statistically significant improvement in speech perception in noise (with speech and noise coming from the front) at 3 and 6 months follow-up compared to baseline (**Fig 2**). Also the BCD group had a significant improvement compared to baseline, but only at 6 months follow-up. Even when one outlier was not included in the analysis, these results remained statistically significant. There were no significant changes for the CROS and No treatment groups.

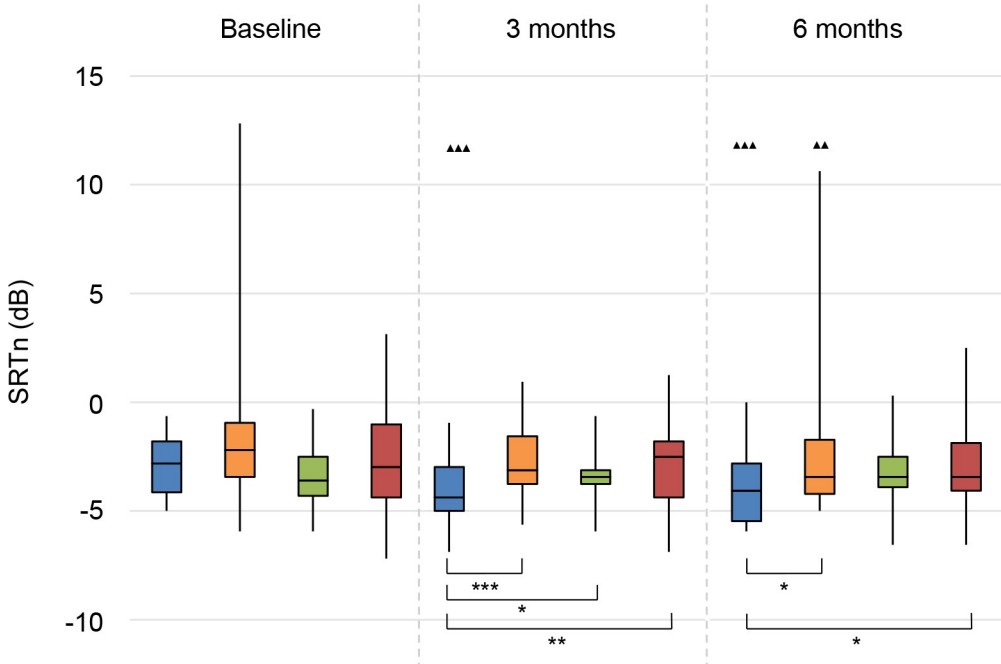

**Fig 2. Speech perception in noise.** Speech reception threshold in noise (SRTn) in decibels (dB), configuration $S_0N_0$. Statistical difference compared to baseline. ▲▲▲ = $p < 0.001$; ▲▲ = $p < 0.01$; ▲ = $p < 0.05$. Statistical difference compared to 3 months follow-up. ■■■ = $p < 0.001$; ■■ = $p < 0.01$; ■ = $p < 0.05$. Statistical difference between groups at same follow-up moment. *** = $p < 0.001$; ** = $p < 0.01$; * = $p < 0.05$.

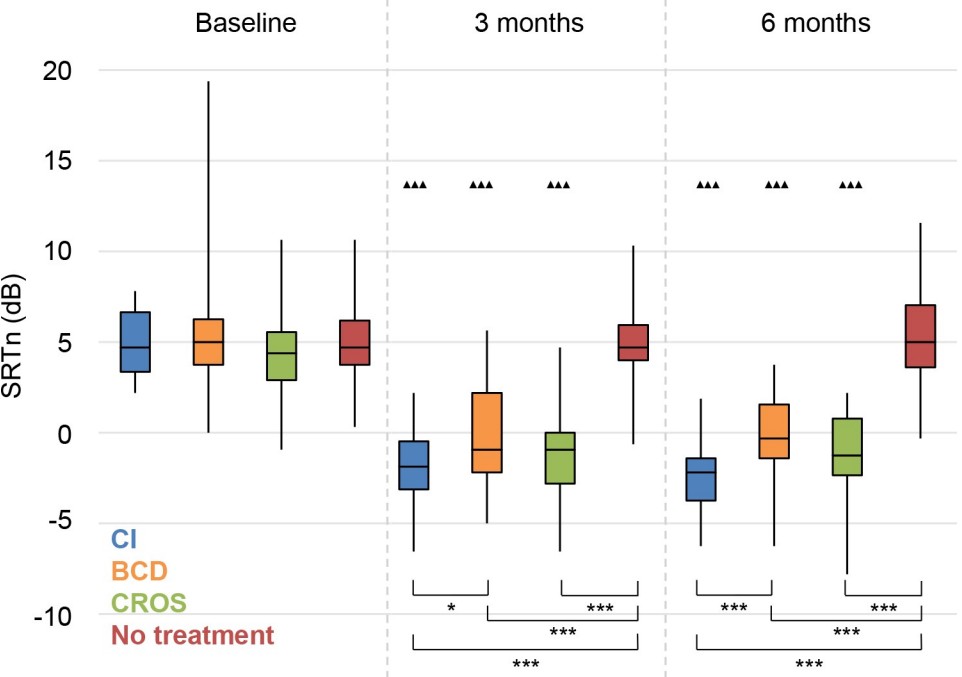

**Fig 3. Speech perception in noise.** Speech reception threshold in noise (SRTn) in decibels (dB), configuration $S_{pe}N_{be}$, Statistical difference compared to baseline. ▲▲▲ = $p < 0.001$; ▲▲ = $p < 0.01$; ▲ = $p < 0.05$. Statistical difference compared to 3 months follow-up. ■■■ = $p < 0.001$; ■■ = $p < 0.01$; ■ = $p < 0.05$. Statistical difference between groups at same follow-up moment. *** = $p < 0.001$; ** = $p < 0.01$; * = $p < 0.05$.

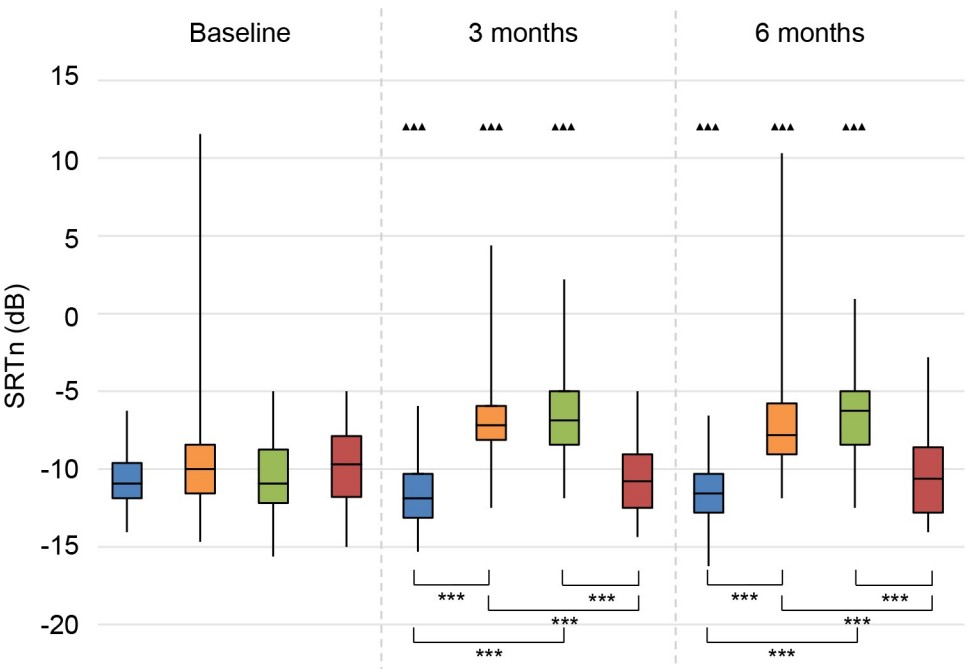

**Fig 4. Speech perception in noise.** Speech reception threshold in noise (SRTn) in decibels (dB), configuration $S_{be}N_{pe}$. Statistical difference compared to baseline. ▲▲▲ = $p < 0.001$; ▲▲ = $p < 0.01$; ▲ = $p < 0.05$. Statistical difference compared to 3 months follow-up. ■■■ = $p < 0.001$; ■■ = $p < 0.01$; ■ = $p < 0.05$. Statistical difference between groups at same follow-up moment. *** = $p < 0.001$; ** = $p < 0.01$; * = $p < 0.05$.

At 3 months follow-up, the CI group performed better than all other groups. At 6 months, the CI group performed significantly better than the BCD and No treatment groups. The difference between the CI group and the CROS group was not statistically significant.

$S_{pe}N_{be}$. With speech directed to the poor ear, all treatment groups (CI, BCD, CROS) showed statistically significantly improved speech perception at 3 and 6 months follow-up compared to baseline (**Fig 3**). There was no significant change for the No treatment group during follow-up.

At 3 and 6 months follow-up, the patients in all treatment groups had better speech perception in noise than the patients in the No treatment group. The CI group also performed statistically significantly better than the BCD group.

$S_{be}N_{pe}$. With speech directed to the better ear, the CI group showed a significant improvement in speech perception at 3 and 6 months follow-up compared to baseline, whereas both the BCD and CROS group deteriorated significantly (**Fig 4**). The No treatment group showed no significant changes over time.

At 3 and 6 months follow-up, the BCD and CROS group had a significantly worse speech perception than the CI and No treatment groups.

## Sound localisation

See **Figs 5–7** for the results on sound localisation. See **S2 Table** for exact data. In all configurations and in all groups, a wide range in performance was observed.

The sound localisation of the CI group significantly improved between baseline and 3 and 6 months follow-up in all configurations. For the 15° and 60° configurations, there was also a statistically significant improvement between 3 and 6 months follow-up. There was no difference compared to baseline for the other groups.

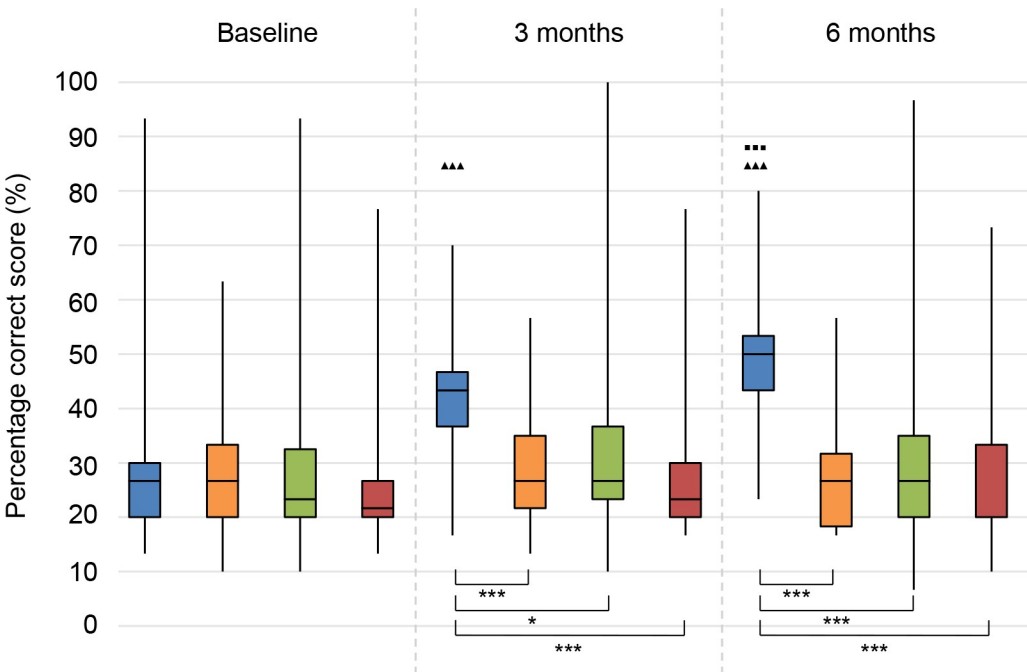

**Fig 5. Sound localisation.** Sound localisation as a percentage correct score (%) for configuration 15˚ angle between 5 loudspeakers (chance level: 20%). Statistical difference compared to baseline. ▲▲▲ = $p < 0.001$; ▲▲ = $p < 0.01$; ▲ = $p < 0.05$. Statistical difference compared to 3 months follow-up. ■■■ = $p < 0.001$; ■■ = $p < 0.01$; ■ = $p < 0.05$. Statistical difference between groups at same follow-up moment. *** = $p < 0.001$; ** = $p < 0.01$; * = $p < 0.05$.

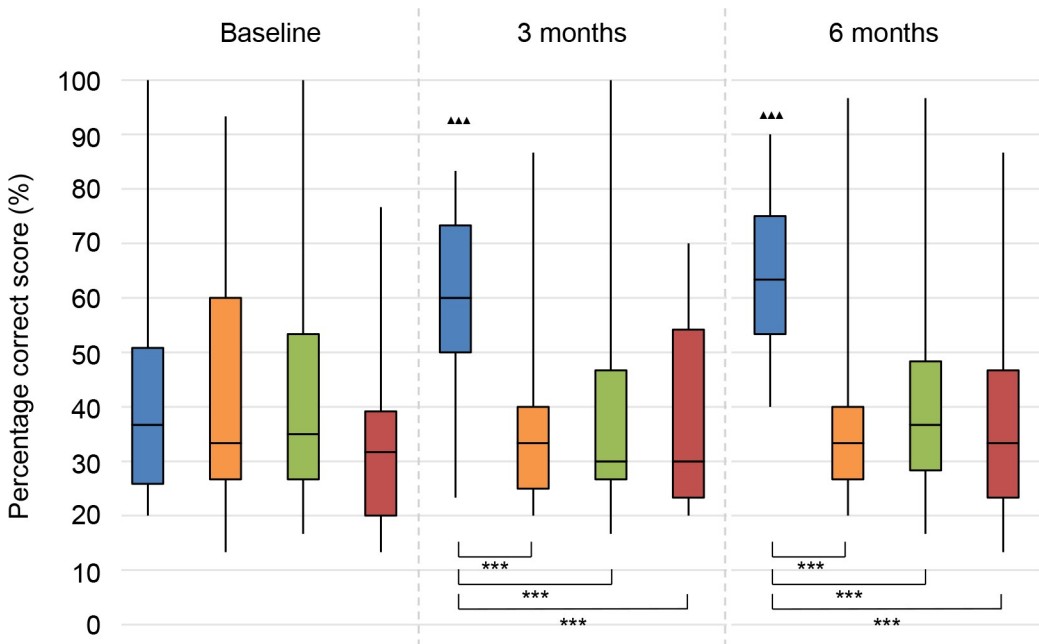

**Fig 6. Sound localisation.** Sound localisation as a percentage correct score (%) for configuration 30˚ angle between 5 loudspeakers (chance level: 20%), Statistical difference compared to baseline. ▲▲▲ = $p < 0.001$; ▲▲ = $p < 0.01$; ▲ = $p < 0.05$. Statistical difference compared to 3 months follow-up. ■■■ = $p < 0.001$; ■■ = $p < 0.01$; ■ = $p < 0.05$. Statistical difference between groups at same follow-up moment. *** = $p < 0.001$; ** = $p < 0.01$; * = $p < 0.05$.

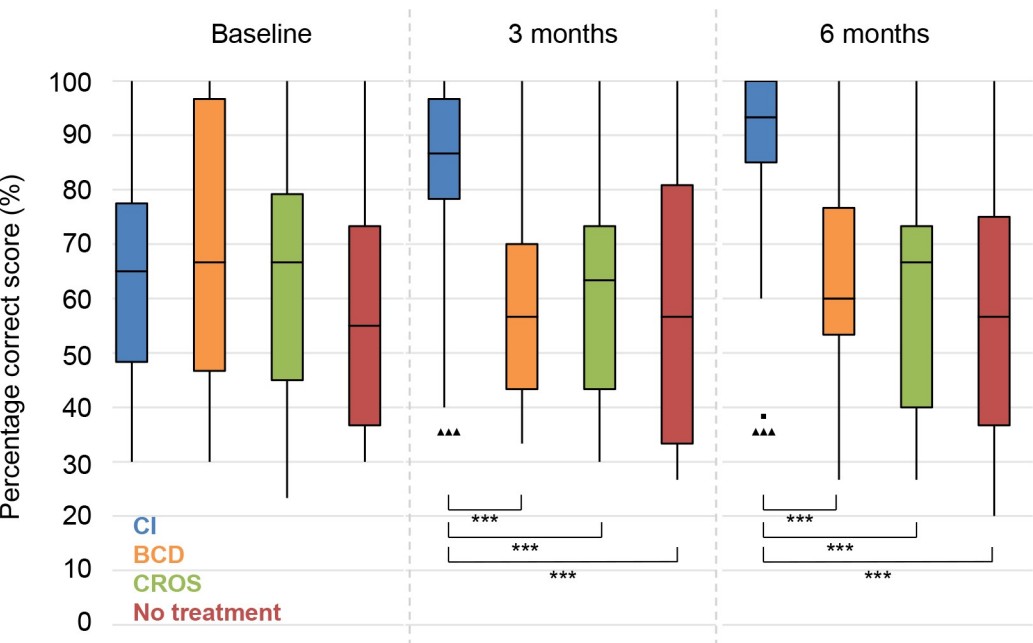

**Fig 7. Sound localisation.** Sound localisation as a percentage correct score (%) for configuration 60˚ angle between 3 loudspeakers (chance level: 33.3%). Statistical difference compared to baseline. ▲▲▲ = $p < 0.001$; ▲▲ = $p < 0.01$; ▲ = $p < 0.05$. Statistical difference compared to 3 months follow-up. ■■■ = $p < 0.001$; ■■ = $p < 0.01$; ■ = $p < 0.05$. Statistical difference between groups at same follow-up moment. *** = $p < 0.001$; ** = $p < 0.01$; * = $p < 0.05$.

At both 3 and 6 months follow-up, the CI group performed significantly better than all other groups. There were no statistically significant differences between the other groups.

## Tinnitus

At all follow-up moments and in all groups, there were some patients who indicated that they did not suffer from tinnitus (anymore) (**Table 2**). Since these patients had no tinnitus, they did not complete the TQ and THI at that moment. Patients with no tinnitus at baseline did not develop tinnitus after any intervention. See **S2 Table** for exact data on the TQ and THI.

**TQ.** There was a significant decrease in tinnitus-related distress for the CI group and the BCD group at 3 and 6 months follow-up compared to baseline (**Fig 8**).

At 3 and 6 months follow-up, the CI group had significantly lower TQ scores than the No treatment group.

**THI.** We measured a significant decrease in impact of tinnitus on daily life for the CI group at 3 and 6 months follow-up compared to baseline (**Fig 9**).

Again, at 3 and 6 months follow-up, the CI group had significantly lower THI scores than the No treatment group.

## Disease-specific QoL

See **S2 Table** for exact data on the disease-specific QoL questionnaires.

**SSQ.** *Speech-hearing*. There was a significant improvement (higher score) for all treatment groups at 3 and 6 months follow-up compared to baseline (**Fig 10**). There was no significant change compared to baseline for the No treatment group.

At 3 and 6 months follow-up, the CI group had significantly higher median values compared to all other groups, and both the BCD and CROS groups scored better than the No treatment group.

**Table 2. Number of patients with and without tinnitus per group per follow-up moment.**

| Tinnitus? | CI | | | BCD | | | CROS | | | No treatment | | |
|---|---|---|---|---|---|---|---|---|---|---|---|---|
| | Baseline | 3 months | 6 months | Baseline | 3 months | 6 months | Baseline | 3 months | 6 months | Baseline | 3 months | 6 months |
| Yes | 26 | 20 | 21 | 22 | 18 | 21 | 33 | 31 | 28 | 24 | 21 | 24 |
| No | 2 | 5 | 5 | 3 | 3 | 2 | 1 | 1 | 2 | 2 | 3 | 3 |
| *Missings* | 0 | 2 | 1 | 0 | 2 | 0 | 0 | 2 | 1 | 0 | 2 | 0 |
| **Total number** | 28 | 27 | 27 | 25 | 23 | 23 | 34 | 34 | 31 | 26 | 26 | 27 |

In the row 'total number', the total number of patients per group and per follow-up moment is mentioned, which corresponds to the numbers shown in the Flow Diagram, **Fig 1**.

In the CI group, two patients did not suffer from tinnitus at baseline and they remained tinnitus-free after 3 and 6 months follow-up. Three patients reported to have no tinnitus anymore after cochlear implantation at 3 and 6 months follow-up.

In the BCD group, there were three patients who did not suffer from tinnitus at baseline. One of them was a patient who switched to the No treatment group after 3 and 6 months follow-up. The two other patients also did not suffer from tinnitus at 3 and 6 months follow-up. One other patient who suffered from tinnitus at baseline reported not to have tinnitus at 3 months follow-up, but the tinnitus had returned at 6 months follow-up.

In the CROS group, one patient had no tinnitus at baseline, 3 and 6 months follow-up. One other patient reported to have no tinnitus anymore at 6 months follow-up, whereas tinnitus was present at baseline and 3 months follow-up.

In the No treatment group, two patients were tinnitus-free at baseline, 3 and 6 months follow-up. The aforementioned patient with no tinnitus who switched from the BCD group was analysed in the No treatment group at 3 and 6 months follow-up.

Abbreviations: CI = Cochlear Implant, BCD = Bone Conduction Device, CROS = Contralateral Routing of Sound hearing aid, SD = standard deviation.

*Spatial hearing.* We observed a significant improvement for all treatment groups at 3 months follow-up, and for CI and BCD groups also at 6 months follow-up (**Fig 11**). The score for the CROS group slightly decreased at 6 months follow-up and was therefore not

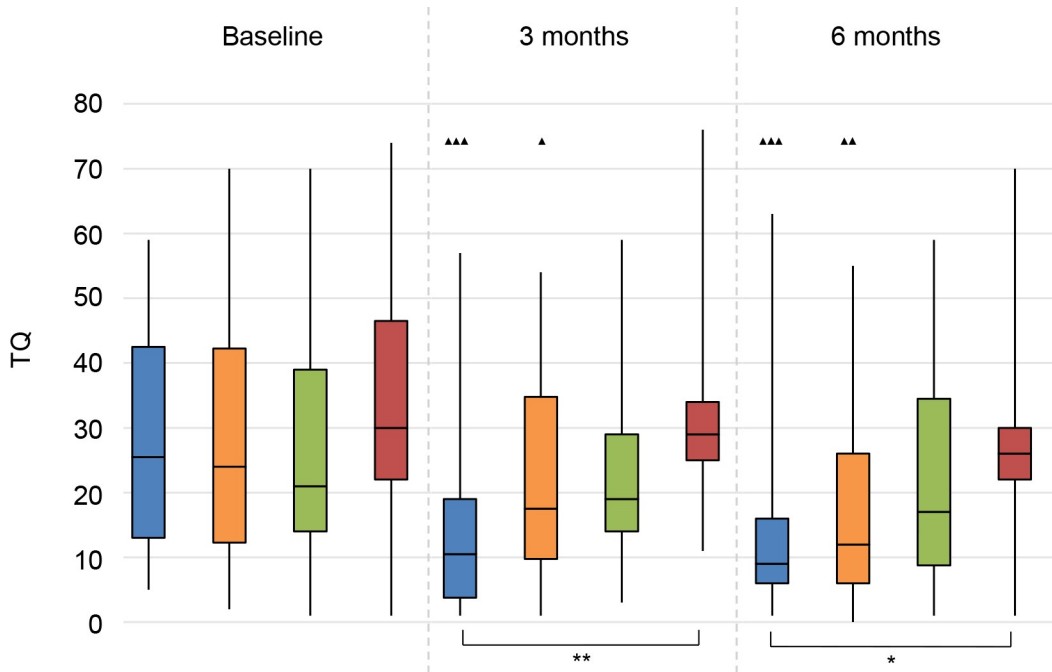

**Fig 8. Tinnitus Questionnaire (TQ).** Statistical difference compared to baseline. ▲▲▲ = $p < 0.001$; ▲▲ = $p < 0.01$; ▲ = $p < 0.05$. Statistical difference compared to 3 months follow-up. ■■■ = $p < 0.001$; ■■ = $p < 0.01$; ■ = $p < 0.05$. Statistical difference between groups at same follow-up moment. *** = $p < 0.001$; ** = $p < 0.01$; * = $p < 0.05$.

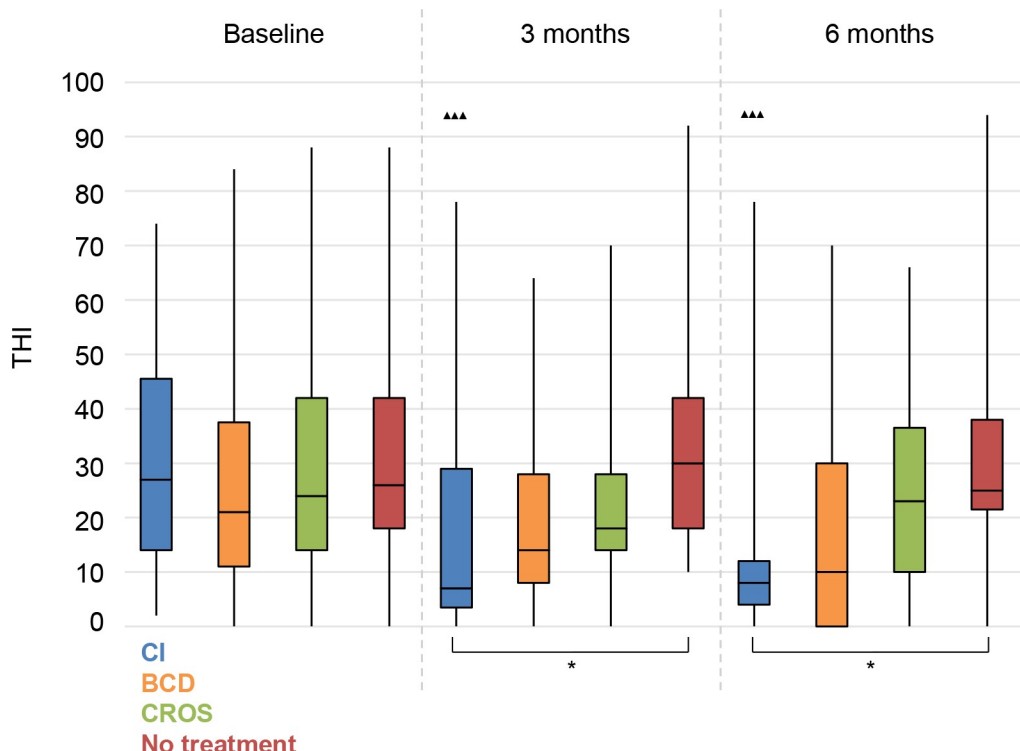

**Fig 9. Tinnitus Handicap Inventory (THI).** Statistical difference compared to baseline. ▲▲▲ = $p < 0.001$; ▲▲ = $p < 0.01$; ▲ = $p < 0.05$. Statistical difference compared to 3 months follow-up. ■■■ = $p < 0.001$; ■■ = $p < 0.01$; ■ = $p < 0.05$. Statistical difference between groups at same follow-up moment. *** = $p < 0.001$; ** = $p < 0.01$; * = $p < 0.05$.

significantly improved compared to baseline. There was no significant change for the No treatment group.

At 3 and 6 months follow-up, the scores for the CI group were significantly higher compared to all other groups. Also, at 6 months follow-up, the BCD group reached higher scores than the No treatment group.

*Qualities of hearing.* For all treatment groups, the scores at 3 months follow-up were significantly better than the baseline scores (**Fig 12**). At 6 months follow-up, this improvement remained for only the CI group. For the BCD group, there was a significant deterioration between 3 and 6 months follow-up. There was no significant change for the No treatment group.

At 3 months follow-up, the scores of all treatment groups were significantly better than the scores of the No treatment group. At 6 months follow-up, the scores of only the CI and CROS groups were significantly better than the scores of the No treatment group.

**APHAB.** *Ease of Communication (EC).* At 3 and 6 months follow-up, all treatment groups had an improvement (i.e. a decrease in APHAB score) on this subscale (**Fig 13**).

At 3 and 6 months, all treatment groups had better scores than the No treatment group. The CI group had significantly better scores than the CROS group.

*Background noise (BN).* All treatment groups had significantly improved scores at 3 and 6 months follow-up (**Fig 14**).

All treatment groups scored significantly better than the No treatment group at 3 and 6 months follow-up. The CI group had significantly better scores than the CROS group at 3 and 6 months follow-up, and also significantly better scores than the BCD group at 6 months follow-up.

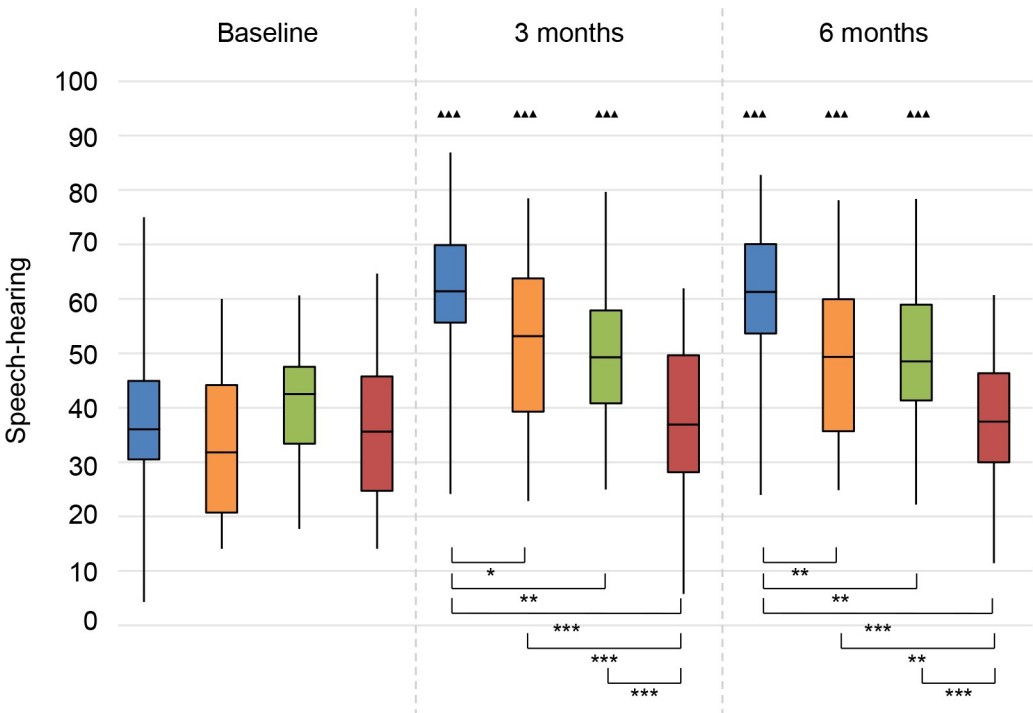

**Fig 10. Speech, Spatial and Qualities of hearing questionnaire (SSQ).** Speech-hearing subscale. Statistical difference compared to baseline. ▲▲▲ = $p < 0.001$; ▲▲ = $p < 0.01$; ▲ = $p < 0.05$. Statistical difference compared to 3 months follow-up. ■■■ = $p < 0.001$; ■■ = $p < 0.01$; ■ = $p < 0.05$. Statistical difference between groups at same follow-up moment. *** = $p < 0.001$; ** = $p < 0.01$; * = $p < 0.05$.

*Reverberant conditions (RV)*. The scores of all treatment groups had significantly improved at 3 and 6 months follow-up (**Fig 15**). However, the scores for the CROS group significantly deteriorated between 3 and 6 months follow-up.

At 3 months follow-up, the CI group and the CROS group scored significantly better than the No treatment group. The CROS group also scored better than the BCD group. At 6 months follow-up, the CI group outperformed all other groups. All treatment groups scored better than the No treatment group.

*Aversiveness of sounds (AS)*. At 3 months follow-up, the scores of all treatment groups were better than the score of the No treatment group (**Fig 16**). This benefit remained for the BCD and CI groups, whereas this benefit decreased for the CROS group, leading to a non-significant difference between baseline and 6 months follow-up for the CROS group.

At 3 months of follow-up, all treatment groups scored better than the No treatment group. At 6 months follow-up, the CI group scored significantly better than the CROS and No treatment groups. The scores of the BCD group were significantly better than the scores of the No treatment group.

**GBI.** On the *general* subscale, the scores improved (i.e. >0) in all treatment groups. The CI group had a significantly better score than the CROS group at 3 months follow-up. Since the *social support* subscale and the *physical health* subscale of the GBI are calculated by adding the responses of three questions each, there was hardly any variability in the data (quartile 1, the median, and quartile 3 had exactly the same values; see **S2 Table**).

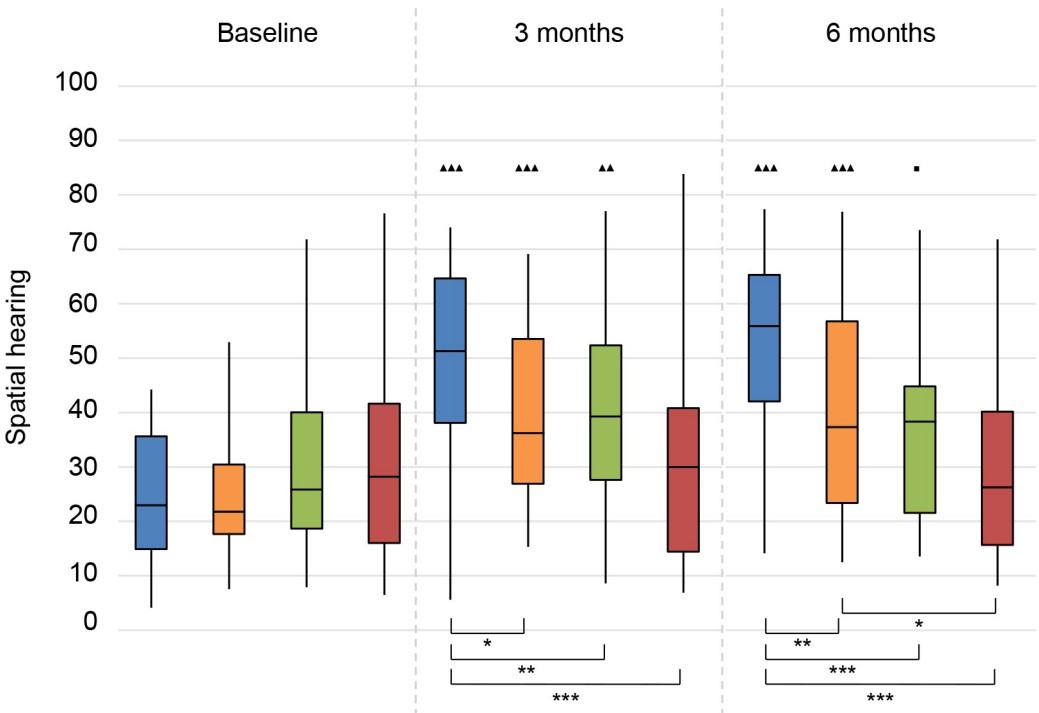

**Fig 11. Speech, Spatial and Qualities of hearing questionnaire (SSQ).** Spatial hearing subscale. Statistical difference compared to baseline. ▲▲▲ = *p* < 0.001; ▲▲ = *p* < 0.01; ▲ = *p* < 0.05. Statistical difference compared to 3 months follow-up. ■■■ = *p* < 0.001; ■■ = *p* < 0.01; ■ = *p* < 0.05. Statistical difference between groups at same follow-up moment. *** = *p* < 0.001; ** = *p* < 0.01; * = *p* < 0.05.

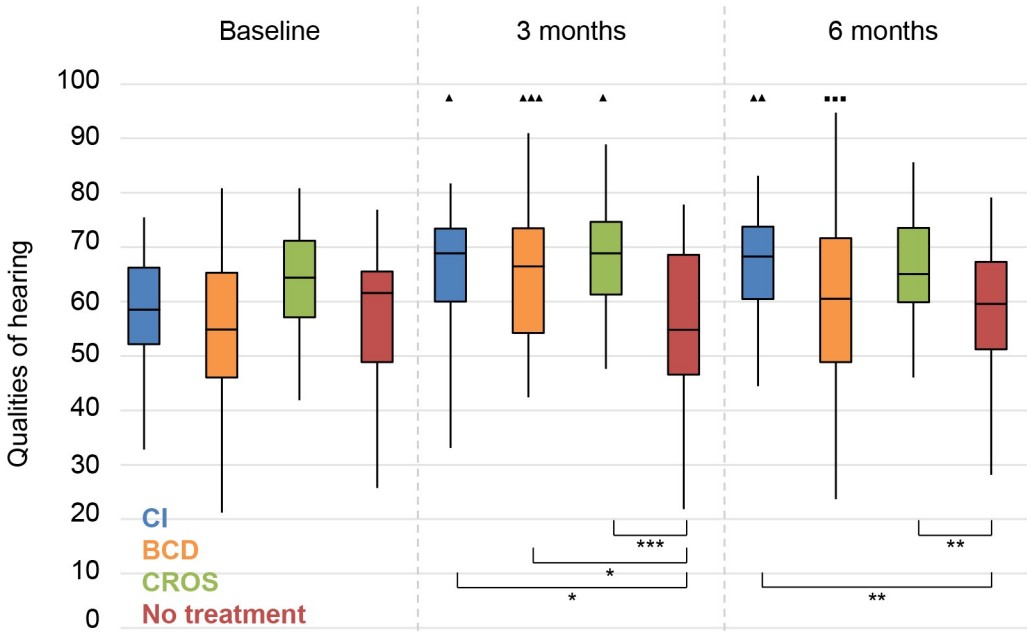

**Fig 12. Speech, Spatial and Qualities of hearing questionnaire (SSQ).** Qualities of Hearing subscale. Statistical difference compared to baseline. ▲▲▲ = *p* < 0.001; ▲▲ = *p* < 0.01; ▲ = *p* < 0.05. Statistical difference compared to 3 months follow-up. ■■■ = *p* < 0.001; ■■ = *p* < 0.01; ■ = *p* < 0.05. Statistical difference between groups at same follow-up moment. *** = *p* < 0.001; ** = *p* < 0.01; * = *p* < 0.05.

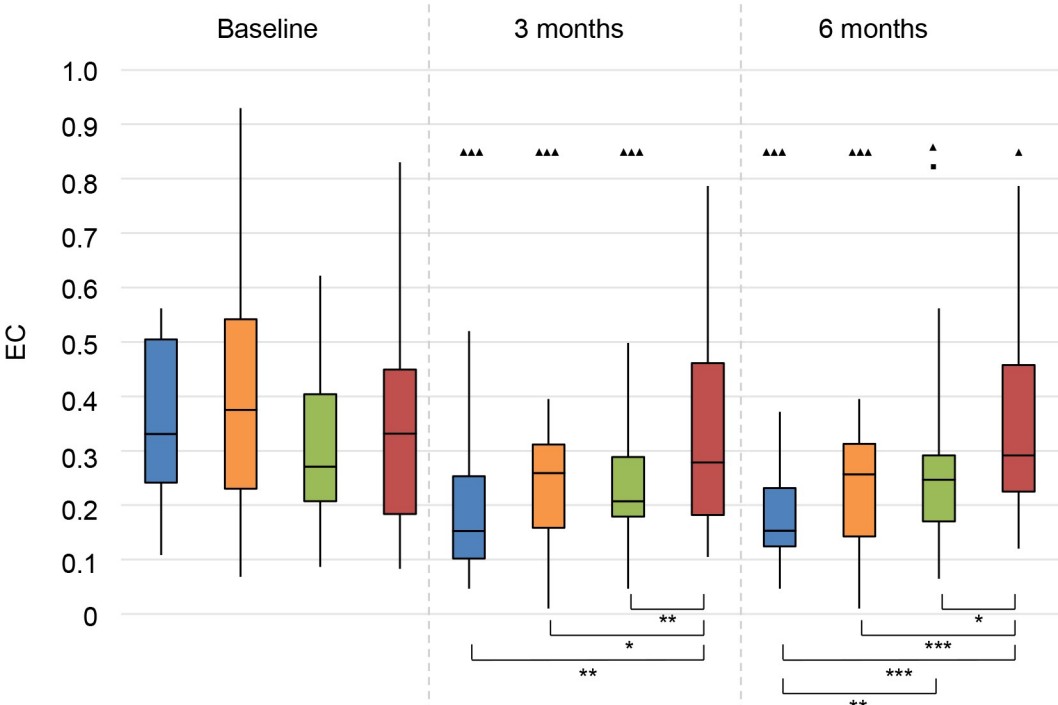

**Fig 13. Abbreviated Profile for Hearing Aid Benefit questionnaire (APHAB).** Ease of communication (EC). Statistical difference compared to baseline. ▲▲▲ = $p < 0.001$; ▲▲ = $p < 0.01$; ▲ = $p < 0.05$. Statistical difference compared to 3 months follow-up. ■■■ = $p < 0.001$; ■■ = $p < 0.01$; ■ = $p < 0.05$. Statistical difference between groups at same follow-up moment. *** = $p < 0.001$; ** = $p < 0.01$; * = $p < 0.05$.

## Serious Adverse Events (SAEs)

There were two related SAEs in the BCD group: two implant extrusions occurred; one patient was re-implanted, in the other case no re-implantation was performed at the patient's request.

There were also unrelated SAEs: in the CI group one patient suffered from a transient ischemic attack several months after implantation; in the CROS group, one patient had a myocardial infarction for which he underwent percutaneous coronary intervention, and one patient had nasal polyps for which he underwent sinus surgery; in the No treatment group one patient had an arm fracture requiring surgery.

## Discussion

We presented the first results of this RCT with 120 included SSD patients evaluating CI, BCD, CROS, and No treatment at 3 and 6 months follow-up. The randomised allocation ensured equal distribution of known and unknown confounders amongst groups.

### Speech perception in noise

On our primary outcome ($S_0N_0$), the CI group demonstrated a significant improvement compared to baseline, which can be attributed to the summation effect. Other studies did not measure an advantage in this configuration after 6 months CI use in SSD patients [14,31]. In the $S_{pe}N_{be}$ configuration, all treatment options led to an improved speech perception in noise, since the sound from the side of the poor ear is delivered to the better ear. However, in the $S_{be}N_{pe}$ configuration the opposite happened: the BCD and CROS

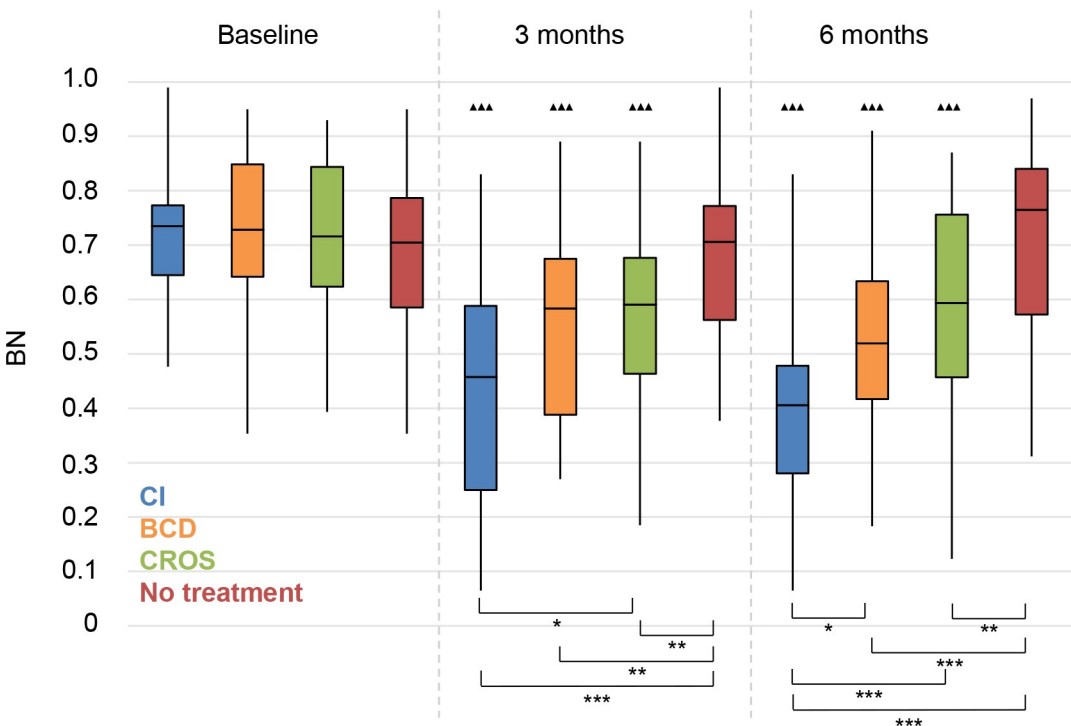

**Fig 14. Abbreviated Profile for Hearing Aid Benefit questionnaire (APHAB).** Background noise (BN). Statistical difference compared to baseline. ▲▲▲ = $p < 0.001$; ▲▲ = $p < 0.01$; ▲ = $p < 0.05$. Statistical difference compared to 3 months follow-up. ■■■ = $p < 0.001$; ■■ = $p < 0.01$; ■ = $p < 0.05$. Statistical difference between groups at same follow-up moment. *** = $p < 0.001$; ** = $p < 0.01$; * = $p < 0.05$.

negated the positive effect of the head shadow as the noise on the poor side was sent to the better ear and thus impeded speech intelligibility. In this $S_{be}N_{pe}$ configuration, the BCD and CROS patients performed worse than at baseline, and worse than patients in the No treatment group at 3 and 6 months follow-up. In contrast to the BCD and CROS groups, there was no disadvantage for the CI patients, proving that the squelch effect is present, as was objectified before after cochlear implantation in patients with asymmetrical hearing loss [12,32].

The speech perception in noise may have been influenced by the hearing thresholds in the better ear. However, there were no significant differences in $PTA_{be}$ between groups after treatment allocation (see **S1 Table**). Noticeably, the speech perception in noise of the CI group was not as good as the speech perception in noise of normal-hearing subjects [24].

## Sound localisation

Cochlear implantation led to a significant improvement in sound localisation in multiple configurations, which is in line with literature [9,12–14,17,20]. The benefit of binaural input is present very early after CI activation, as was described by others [33]. We found that the sound localisation ability of the CI patients further improved from 3 to 6 months follow-up for two configurations (15° and 60°), which is also in line with what other researchers have described [13,14,34]. Nevertheless, sound localisation is still harder for SSD patients with a CI than for normal-hearing subjects [24,35].

In line with previous articles, we did not find improvement of sound localisation ability with BCD or CROS [11,36,37].

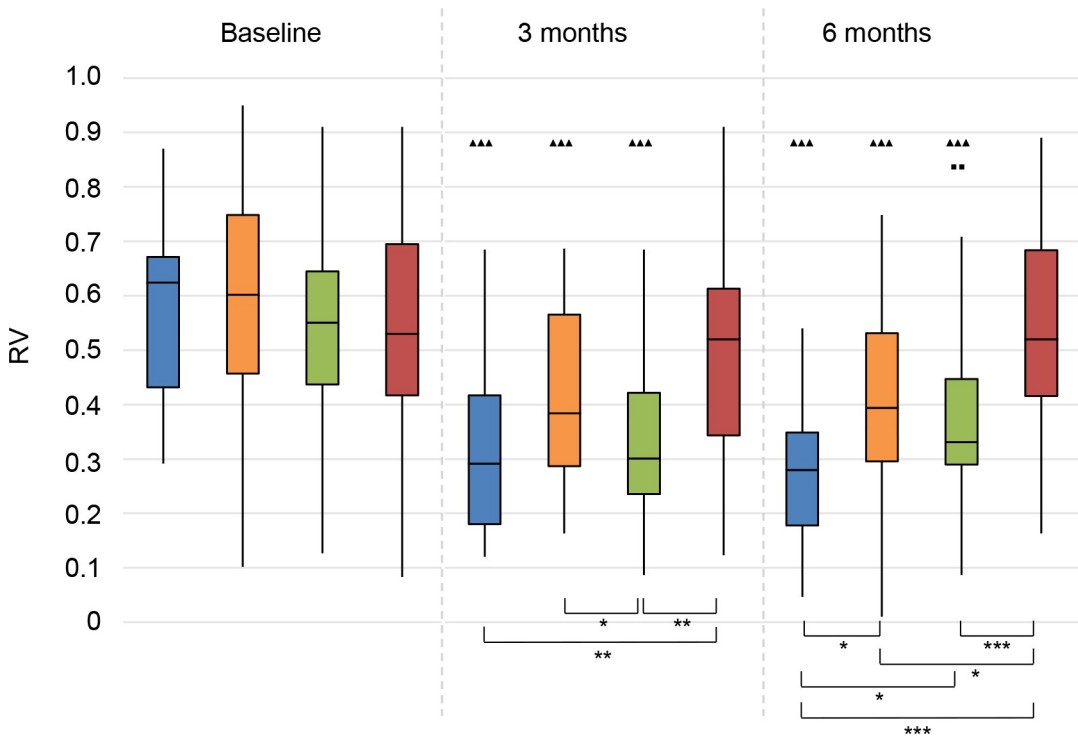

**Fig 15. Abbreviated Profile for Hearing Aid Benefit questionnaire (APHAB).** Reverberation (RV). Statistical difference compared to baseline. ▲▲▲ = $p < 0.001$; ▲▲ = $p < 0.01$; ▲ = $p < 0.05$. Statistical difference compared to 3 months follow-up. ■■■ = $p < 0.001$; ■■ = $p < 0.01$; ■ = $p < 0.05$. Statistical difference between groups at same follow-up moment. *** = $p < 0.001$; ** = $p < 0.01$; * = $p < 0.05$.

## Tinnitus

For the first time in 2008, Van de Heyning et al. [5] reported a significant decrease in tinnitus distress (TQ) and tinnitus loudness (Visual Analogue Scale (VAS) score) following cochlear implantation in unilaterally deaf patients suffering from incapacitating tinnitus. Later, others reported beneficial results of electrical stimulation by cochlear implantation on tinnitus as well [17,20,31,38,39]. An important issue to be mentioned is the absence of a minimum level of tinnitus distress to be eligible for inclusion in our study. Other study groups often required a minimum level of tinnitus distress, such as a minimum VAS score on tinnitus loudness of 6 out of 10 [5,12,32] or a minimum THI score of 58 [40,41]. This requirement explains their higher baseline values for tinnitus distress. Despite our lower baseline values, we also objectified a significant decrease in tinnitus distress after 3 and 6 months for the CI patients.

Interestingly, we identified a significant reduction on tinnitus distress after BCD implantation with the TQ, without a significant change in tinnitus-related effects on daily life as measured by the THI. One other study found a reduction of the impact of tinnitus on daily life after BCD [42]. The mechanism is unclear, but stimulation of the contralateral auditory pathway may play an important role in the experienced tinnitus suppression [42].

Some CI patients reported reduction of tinnitus as the most important advantage of their CI, rather than improved speech perception in noise or sound localisation. This finding could be an important advantage that should be mentioned when counselling SSD patients on expected outcomes of CI.

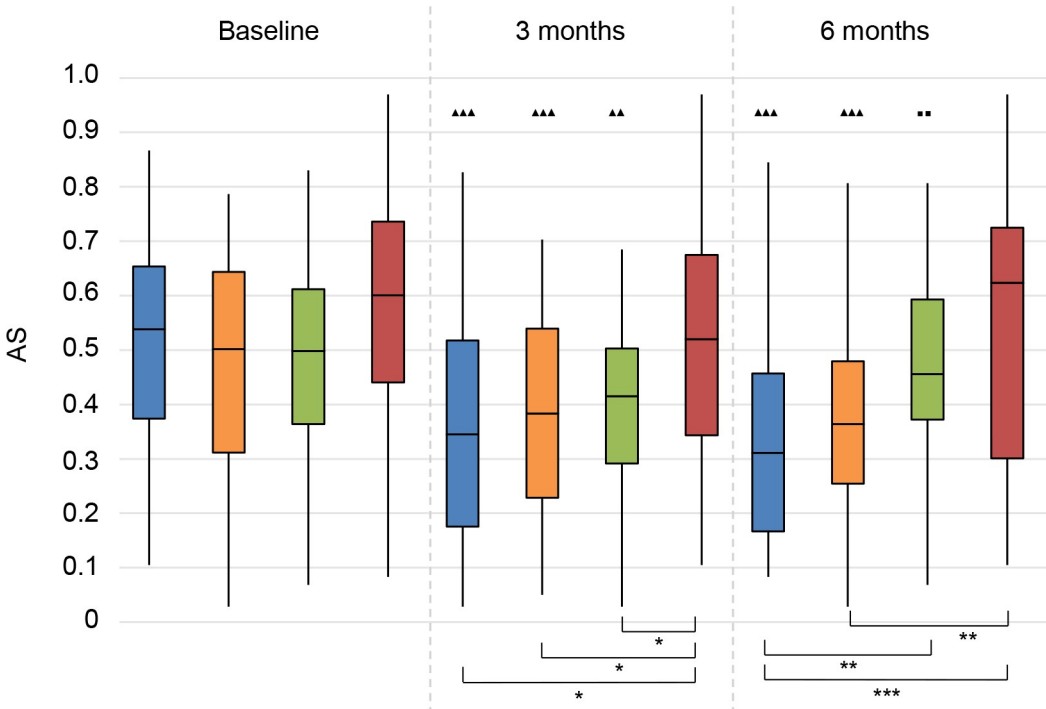

**Fig 16. Abbreviated Profile for Hearing Aid Benefit questionnaire (APHAB).** Aversiveness of sounds (AS). Statistical difference compared to baseline. ▲▲▲ = $p < 0.001$; ▲▲ = $p < 0.01$; ▲ = $p < 0.05$. Statistical difference compared to 3 months follow-up. ■■■ = $p < 0.001$; ■■ = $p < 0.01$; ■ = $p < 0.05$. Statistical difference between groups at same follow-up moment. *** = $p < 0.001$; ** = $p < 0.01$; * = $p < 0.05$.

### Disease-specific QoL

For most subscales on the disease-specific QoL questionnaires, there was an improvement for all treatment groups compared to baseline, and compared to the No treatment group at 3 and 6 months follow-up. Our SSQ results are in line with the results of studies that also used the SSQ questionnaire in SSD patients after cochlear implantation [31,38].

Although the APHAB questionnaire has not been widely used in reports on cochlear implantation for SSD, we implemented this questionnaire because it has been extensively used to evaluate BCD and CROS [11]. Also on the APHAB questionnaire we found an improvement for all treatment groups on most subscales, which equals findings by others [11,43].

### Trial period

The proportion of patients opting for a BCD after the trial period in our study (29.4%) was on the low end of the range described in literature for patients opting for a BCD after a BCD trial period (32.0–69.6%) [44]. Patients in our study were offered a cross-over design which enabled them to test two devices, which may explain the lower proportion of patients opting for a BCD after the trial period. Moreover, the proportion of patients opting for a BCD could have been influenced by the type of BCD offered during the trial period; we noticed that the satisfaction with the newer Baha[®] 5 Power was significantly higher than with the older Baha[®] BP110 (45.5% versus 19.2% opted for BCD after the trial period, respectively). Noticeably, 40.0% of patients opted for CROS. This high proportion may be attributed to recent advances in (directional) microphone technology and signal processing algorithms. Importantly, a considerable proportion of patients was satisfied with either modality and preferred these treatments over no treatment.

Other studies have also provided a trial with BCD or CROS to SSD patients. Though, due to lack of reported outcomes in these studies, comparison with our results is hindered. For example, Arndt et al. [31] provided a CI to 11 SSD patients that were not satisfied with the effects of BCD or CROS in a trial period. However, the proportion of patients that opted for BCD or CROS after the trial period was not mentioned, and no objective outcomes of hearing performance were published from these satisfied patients. Likewise, SSD patients were offered a CI in a Swiss study if they had unsatisfactory benefit from a trial with CROS or BCD: forty-four of 54 study patients did not proceed to cochlear implantation [39]. Unfortunately, there were no results available from those patients who opted for BCD or CROS. Of the ten patients that were implanted, only results with CI were published and not from the trial period with CROS or BCD [39]. Kurz et al. [14] implemented the study design that was proposed at an international consensus meeting [1], and found that 16 of 34 patients decided not to opt for cochlear implantation after a trial period with BCD and CROS (3 weeks each).

As an advantage over studies with a trial period of BCD and CROS before the option to choose for CI, we were able to make a longitudinal between-group comparison comparing CI, BCD, CROS, and no treatment. With our study design we diminished the influence of the opinion of patients on CI outcome by previous BCD or CROS treatments, and also diminished the influence on patients' opinions on BCD and CROS. With a more extended analysis of the trial period in the future, we hope to identify prognostic factors that may predict if BCD or CROS are successful in specific categories of patients. Additionally, we will also analyse the disease-specific QoL questionnaires that were completed after each 6-week phase of the trial period (SSQ, APHAB, and GBI).

## Strengths

There are several strengths that characterize our trial. Firstly, we randomly allocated patients to intervention groups. Our random allocation ensured equal distribution of patient characteristics across groups, thereby eliminating the influence of differences in these characteristics on the outcomes [45,46]. Secondly, we compared CI to the treatment options for patients with SSD currently available in many countries. Thirdly, our design provided not only a within-subject comparison, but also allowed a between-group comparison. It also enabled us to evaluate the performance following SSD in the No treatment group. Finally, we had very few missing data permitting a complete case-analysis.

## Limitations

Our study also has limitations. First, following the stringent in- and exclusion criteria, we included only a subset of patients with SSD, which may limit the generalisability of our results. Moreover, our study patients were highly motivated to return for follow-up visits, whereas in real-life compliance may not be as high. Possibly, non-use of initially started treatment may occur when follow-up time is longer [19,34,47]. We therefore implemented a follow-up period of five years. Second, neither patients nor personnel were blinded for the allocated intervention. Given the nature of the interventions, blinding was impossible to achieve, which may have influenced outcomes; this is a well-known problem in non-pharmacological studies [48]. Moreover, the magnitude of the placebo effect in surgical trials is unknown [49]. Third, we used fixed block sizes (n = 8, 2:3:3) in our block randomisation, which may result in a predictable allocation of participants based on previous randomisations. This lack of allocation concealment may lead to selection bias [50]. However, the baseline characteristics showed no statistically significant differences between allocation groups. Fourth, due to the long inclusion period of our trial, we implemented newly developed devices in the trial. Consequently, not all

patients tested the same BCD in the trial period, or had the same type of CROS or CI fitted. We feel this is part of conducting a multi-annual trial and reflects technological advancements similar to clinical care settings. Because of standardised fitting strategies for all devices, we think the groups are still comparable. Fifth, the TQ and THI are not designed to measure treatment effects [51]. To measure treatment-related changes in tinnitus intrusiveness and severity, the more recently developed Tinnitus Functional Index [52] is recommended nowadays. Nevertheless, the commonly used THI makes comparisons with other studies possible [18]. Furthermore, both the TQ and the THI have high internal consistency, high convergence and discriminant validity and good change sensitivity [53]. Thus, both questionnaires appear appropriate to evaluate the effects of tinnitus treatments [53]. Finally, we only measured the median duration of CI use per day, and not the median duration of BCD and CROS use per day. Data of device use from the BCD and CROS devices would have given us insight in the satisfaction with the devices as mirrored from duration of use per day; unfortunately, this data was unavailable.

## Future perspectives

In this paper we presented the first results of the *CINGLE*-trial [21]. Long-term results will follow. Aside from the reported outcome measures, which show the individual benefits, analysis of the costs of the treatment options is necessary to reveal societal benefits. This information will allow health care committees to perform a cost utility analysis, which will help in the consideration whether or not to reimburse cochlear implantation for patients with SSD. Interestingly, a recent modelling study on patients with SSD found that cochlear implantation may be cost-effective compared with no intervention, but bone-conduction implants were unlikely to be cost-effective [54]. Studies comparing CI to other treatment options for SSD and conducting a cost utility analysis are expected in the near future [55,56]. Finally, others work on developing a minimum set of core outcomes for use in future trials of SSD interventions [57].

## Conclusion

In this RCT, we compared CI, BCD, CROS, and No treatment for patients with SSD. Speech perception in noise improved in all configurations for the CI group, whereas speech perception in noise improved or deteriorated for the BCD and CROS groups depending on the configuration. Sound localisation improved in the CI group only. Patients with no tinnitus at baseline did not develop tinnitus after any intervention. After cochlear implantation, three patients reported that their tinnitus had completely disappeared. On both the TQ and THI, a decrease in tinnitus burden was detected for the CI group. In general, all treatment options improved disease-specific QoL on most subscales of the used questionnaires.

Results from long-term follow-up moments will be presented in the future, as well as an analysis of the BCD/CROS trial period and a more detailed analysis of the tinnitus burden. Finally, cost utility analyses on treatment options for SSD are needed to evaluate if cochlear implantation should be reimbursed.

## Supporting information

**S1 File. Original study protocol CINGLE-trial: Cochlear Implantation for siNGLE-sided deafness, a Randomised controlled trial and economic evaluation.**
(PDF)

**S2 File. CONSORT checklist.**
(PDF)

**S3 File. Explanation of numbers per group at different follow-up moments.**
(PDF)

**S1 Table. Patient characteristics after allocation of treatment.**
(PDF)

**S2 Table. Data tables.**
(PDF)

## Acknowledgments

The authors would like to thank D. Ramekers PhD for his help with the statistical analysis.

## Author Contributions

**Conceptualization:** Jeroen P. M. Peters, Gijsbert A. van Zanten, Wilko Grolman, Adriana L. Smit.

**Data curation:** Jeroen P. M. Peters, Jan A. A. van Heteren, Anne W. Wendrich.

**Formal analysis:** Jeroen P. M. Peters, Jan A. A. van Heteren, Anne W. Wendrich.

**Funding acquisition:** Wilko Grolman.

**Investigation:** Jeroen P. M. Peters, Jan A. A. van Heteren, Anne W. Wendrich.

**Methodology:** Jeroen P. M. Peters, Gijsbert A. van Zanten, Wilko Grolman, Adriana L. Smit.

**Project administration:** Jeroen P. M. Peters, Jan A. A. van Heteren, Anne W. Wendrich.

**Resources:** Wilko Grolman, Robert J. Stokroos, Adriana L. Smit.

**Supervision:** Gijsbert A. van Zanten, Wilko Grolman, Robert J. Stokroos, Adriana L. Smit.

**Visualization:** Jeroen P. M. Peters.

**Writing – original draft:** Jeroen P. M. Peters.

**Writing – review & editing:** Jan A. A. van Heteren, Anne W. Wendrich, Gijsbert A. van Zanten, Wilko Grolman, Robert J. Stokroos, Adriana L. Smit.

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
