## [Decision Letter · Decision Letter 0]

21 Dec 2020

PONE-D-20-34585

Short-term Outcomes of Cochlear Implantation for Single-Sided Deafness compared to Bone Conduction Devices and Contralateral Routing of Sound hearing aids – Results of a Randomised Controlled Trial (CINGLE-trial)

PLOS ONE

Dear Dr. Peters,

Thank you for submitting your manuscript to PLOS ONE. After careful consideration, we feel that it has merit but does not fully meet PLOS ONE’s publication criteria as it currently stands. Therefore, we invite you to submit a revised version of the manuscript that addresses the points raised during the review process.

Please pay careful attention to the review comments, and also please address the following issues.

Why would you use blocked randomization, given its well known fatal flaws?

Clearly the data are not normally distributed.  That could have been determined by appeal to the fact that no data are normally distributed.  Why would you feel the need to test to arrive at this conclusion?

Where are the p-values in Table 1?

We look forward to receiving your revised manuscript.

Kind regards,

Vance Berger

Academic Editor

PLOS ONE

Journal Requirements:

2. Please ensure that all deviations from the original approved protocol are discussed and justified in the manuscript text. In particular, please provide a justification why some questionnaire used to assess quality of life as a secondary outcome are not detailed in the Methodology.

Furthermore, please provide additional details to the inclusion criteria within the Manuscript text. And finally please provide clarification regarding the discrepancy in the start date between the manuscript and the clinical trial registry.

3. Thank you for submitting your clinical trial to PLOS ONE and for providing the name of the registry and the registration number. The information in the registry entry suggests that your trial was registered after patient recruitment began. PLOS ONE strongly encourages authors to register all trials before recruiting the first participant in a study.

i) your reasons for your delay in registering this study (after enrolment of participants started);

ii) confirmation that all related trials are registered by stating: “The authors confirm that all ongoing and related trials for this drug/intervention are registered”.

4. Thank you for stating the following in the Financial Disclosure section:

'This study is partly funded by Cochlear Ltd. as an unrestricted research grant. By research contract, Cochlear Ltd. did not have influence on the study design, data collection, analysis, data interpretation, and publication. The funders had no role in study design, data collection and analysis, decision to publish, or preparation of the manuscript.'

We note that you received funding from a commercial source: Cochlear Ltd

Reviewers' comments:

Reviewer's Responses to Questions

**Comments to the Author**

1. Is the manuscript technically sound, and do the data support the conclusions?

Reviewer #1: No

2. Has the statistical analysis been performed appropriately and rigorously? 

Reviewer #1: No

3. Have the authors made all data underlying the findings in their manuscript fully available?

Reviewer #1: No

4. Is the manuscript presented in an intelligible fashion and written in standard English?

Reviewer #1: Yes

5. Review Comments to the Author

Reviewer #1: The authors aimed to compare different treatments (CI, BCD, CROS and no treatment) to support patients with single-side deafness. The trial is described as randomized controlled trial where the description suggests that it is a patient preference design.

From a methodological point of view, I have some difficulties with the analysis and subsequent conclusion.

First, patients were randomised to CI or to one of the treatment sequences BCF followed by CROS or CROS followed by BCF as in a cross-over design. After this “randomized” run-in period, patients from the cross-over groups were treated based on their preference with BCD, CROS or no treatment. Patients treated initially with CI were further treated with CI without having a choice. These options lead to seven different groups, namely CI; BCF based on BCF/CROS; CROS based on BCF/CROS; No treatment based on BCF/CROS; BCF based on CROS/BCF; CROS based on CROS/BCF and No treatment based on CROS/BCF but were analysed as four (CI, BCF, CROS and no treatment). This approach ignores the effects from the pre-treatment, causes confounding and thus implies bias in the analysis. Comparisons using the non-randomised and confounded groups are unreliable because of the presence of uncontrolled confounders.

One possible option would be to use the randomisation status as a covariate. In general, a patient preference trial should be treated as an observational study with known confounding factors adjusted for in the analysis. This also means that the STROBE checklist is more suitable than CONSORT.

The analysis performed by the authors includes comparison between to measurements moments within one group or between two groups at the same follow-up moment. But what is in fact of interest is the comparison of changes between groups. I would suggest to adapt the analysis accordingly to the above mentioned aspects.

Moreover, in the protocol is stated, that missing values will be imputed using a multiple imputation approach and all analyses will be performed on an intention-to-treat basis. Since a protocol is a binding document, the analysis should follow the agreed details.

For the same reason, a justification for the evaluation of the 3 months measurements should be provided, because in the protocol planned evaluation starts at 6 months after randomisation.

6. PLOS authors have the option to publish the peer review history of their article (what does this mean?). If published, this will include your full peer review and any attached files.

Reviewer #1: No

---

## [Author Response · Author response to Decision Letter 0]

21 Jan 2021

PLEASE ALSO SEE THE UPLOADED WORD FILE REPLY TO REVIEWERS

Reply to Reviewers letter PONE-D-20-34585

Original comments 

>>Reply to Reviewers by authors

>>First of all, we would like to thank the editor and reviewer for their detailed feedback and review of our manuscript. In this letter we respond (in blue) to each point raised by the editor and reviewer and made changes in the manuscript accordingly.

Editor

Please pay careful attention to the review comments, and also please address the following issues.

Why would you use blocked randomization, given its well known fatal flaws?

>>The main reason to use block randomisation was to achieve balanced allocation of participants to the three groups over time. We agree that a well-known disadvantage of block randomization is that the allocation of participants may be predictable and result in selection bias when the study groups are unmasked. However, in this trial we did not (could not) blind researchers or patients given for the intervention. Therefore, this disadvantage of block randomization was not applicable to our trial. 

Clearly the data are not normally distributed. That could have been determined by appeal to the fact that no data are normally distributed. Why would you feel the need to test to arrive at this conclusion?

>>Our statistical analysis plan was designed to first check if data were normally distributed, so we could know which type of tests to use (parametric or non-parametric). We agree that our data are clearly not normally distributed; however, there is no harm to check this statistically. For clarity for the readers, we removed the first sentence of the section ‘statistical analysis’ for easier reading and more clarity.

Where are the p-values in Table 1?

>>Since patients were randomly distributed across groups, statistical comparison between baseline characteristics is not advocated. In short, if there would be a statistically significant difference, it would be based on chance by definition (since randomised allocation occurred). See for example the Editorial: Statistical testing for baseline differences between randomised groups is not meaningful via https://www.nature.com/articles/s41393-018-0203-y)

>>We changed the layout of the manuscript, (figure) legends and supporting information according to the PLOS One style requirements. 

2. Please ensure that all deviations from the original approved protocol are discussed and justified in the manuscript text. In particular, please provide a justification why some questionnaire used to assess quality of life as a secondary outcome are not detailed in the Methodology.

>>We explained deviations from the original protocol in the Introduction (lines 131-132), where we specified that some secondary outcomes will be discussed in a separate manuscript that will follow. Moreover, in the Materials and Methods (line 142-144) we detailed the addition of the extra follow-up moment 3 months after device activation. 

Furthermore, please provide additional details to the inclusion criteria within the Manuscript text. 

>>We added information on the inclusion criteria, so it now equals the exact text from the original research protocol and published study protocol manuscript (lines 154-159).

And finally please provide clarification regarding the discrepancy in the start date between the manuscript and the clinical trial registry.

>>Thank you for this remark. The start date in the clinical trial register was incorrect. We updated the info in the clinical trial register (see https://www.trialregister.nl/trial/4457), and now the start date is the same in the clinical trial register and in the manuscript (start date/first inclusion July 15th, 2014). 

3. Thank you for submitting your clinical trial to PLOS ONE and for providing the name of the registry and the registration number. The information in the registry entry suggests that your trial was registered after patient recruitment began. PLOS ONE strongly encourages authors to register all trials before recruiting the first participant in a study.

>>Please see the info online (https://www.trialregister.nl/trial/4457). We registered our trial online (May 6th, 2014) which was before inclusion of the first patient (July 14th, 2014). 

4. Thank you for stating the following in the Financial Disclosure section:

'This study is partly funded by Cochlear Ltd. as an unrestricted research grant. By research contract, Cochlear Ltd. did not have influence on the study design, data collection, analysis, data interpretation, and publication. The funders had no role in study design, data collection and analysis, decision to publish, or preparation of the manuscript.'

We note that you received funding from a commercial source: Cochlear Ltd

>>We amended the Competing Interests Statement in the manuscript (page 2). 

>>We amended the Competing Interests Statement also in the new cover letter. 

5. Review Comments to the Author

Reviewer #1: The authors aimed to compare different treatments (CI, BCD, CROS and no treatment) to support patients with single-side deafness. The trial is described as randomized controlled trial where the description suggests that it is a patient preference design. From a methodological point of view, I have some difficulties with the analysis and subsequent conclusion.

First, patients were randomised to CI or to one of the treatment sequences BCF followed by CROS or CROS followed by BCF as in a cross-over design. After this “randomized” run-in period, patients from the cross-over groups were treated based on their preference with BCD, CROS or no treatment. Patients treated initially with CI were further treated with CI without having a choice. These options lead to seven different groups, namely CI; BCF based on BCF/CROS; CROS based on BCF/CROS; No treatment based on BCF/CROS; BCF based on CROS/BCF; CROS based on CROS/BCF and No treatment based on CROS/BCF but were analysed as four (CI, BCF, CROS and no treatment). This approach ignores the effects from the pre-treatment, causes confounding and thus implies bias in the analysis. Comparisons using the non-randomised and confounded groups are unreliable because of the presence of uncontrolled confounders.

>>We aimed to compare CI to the current clinical practice in The Netherlands (lines 186-188), which consists of a trial period with BCD and CROS. Patients were randomly assigned to CI or to the ‘clinical practice’ groups. We agree with the reviewer that the order in which the ‘current clinical practice’ devices are tested by participants may be of influence on their decision which they prefer. To correct for this ‘order effect’ (line 179), we designed two groups: ‘first BCD, then CROS’, and ‘first CROS, then BCD’, and both then lead to the final groups: BCD, CROS or No treatment (like in current clinical practice). 

It may indeed be interesting, as the reviewer suggests, to analyse differences between patients opting for BCD after having tested the BCD first compared to having the BCD tested second. As detailed in the study protocol (S1_File), a detailed analysis will follow in a later manuscript. For clarity of the reader, we also added this information to the Discussion section (lines 629-631).

One possible option would be to use the randomisation status as a covariate. In general, a patient preference trial should be treated as an observational study with known confounding factors adjusted for in the analysis. This also means that the STROBE checklist is more suitable than CONSORT.

>>Thank you for your comment. Since we used a randomized study design, we used the CONSORT Statement, which is suitable to assess the randomization (e.g. sequence generation) and allocation process. None of the study types that the STROBE checklist was designed for contains this information. Nevertheless, it would be interesting to use the randomisation status as a covariate in the detailed analysis of the trial period.

The analysis performed by the authors includes comparison between to measurements moments within one group or between two groups at the same follow-up moment. But what is in fact of interest is the comparison of changes between groups. I would suggest to adapt the analysis accordingly to the above mentioned aspects.

>>Thank you for this suggestion. For all outcomes, the test/questionnaire median results from baseline and the 3 and 6 months follow-up moments are displayed in Figures 3-6. These results were statistically compared between groups, as was detailed in the data analysis plan. In these figures, readers can also see the changes between groups. All exact medians, interquartile ranges and minimum/maximum values are provided in S5_Tables. In this manuscript, we did not statistically compare the differences/changes in follow-up moments between groups. 

Moreover, in the protocol is stated, that missing values will be imputed using a multiple imputation approach and all analyses will be performed on an intention-to-treat basis. Since a protocol is a binding document, the analysis should follow the agreed details.

>>We thank the reviewer for this remark. Given the very low proportion of missing data in our study, we chose not to use multiple imputation. When designing the study, we took a larger proportion of missing data into account, for which we aimed to use multiple imputation. Due to the nature of our interventions, subject data were analysed ‘as treated’. To make it clear that we deviated from the original protocol, we added this information to the ‘Statistical analysis’ section. 

For the same reason, a justification for the evaluation of the 3 months measurements should be provided, because in the protocol planned evaluation starts at 6 months after randomisation.

>>It is correct that we added the 3 month follow-up moment after first approval of the study protocol. However, we added this follow-up moment before inclusion of the first patient. In lines 142-145 we detailed this deviation from the original protocol.

---

## [Editor Report · Decision Letter 1]

8 Feb 2021

PONE-D-20-34585R1

Short-term Outcomes of Cochlear Implantation for Single-Sided Deafness compared to Bone Conduction Devices and Contralateral Routing of Sound hearing aids – Results of a Randomised Controlled Trial (CINGLE-trial)

PLOS ONE

Dear Dr. Peters,

Thank you for submitting your manuscript to PLOS ONE. After careful consideration, we feel that it has merit but does not fully meet PLOS ONE’s publication criteria as it currently stands. Therefore, we invite you to submit a revised version of the manuscript that addresses the points raised during the review process.

We look forward to receiving your revised manuscript.

Kind regards,

Vance Berger

Academic Editor

PLOS ONE

Additional Editor Comments (if provided):

I am repeating the earlier comments, and your replies to them, for ease of reference.

Please pay careful attention to the review comments, and also please address the following issues.

Why would you use blocked randomization, given its well known fatal flaws?

>>The main reason to use block randomisation was to achieve balanced allocation of participants to the three groups over time. We agree that a well-known disadvantage of block randomization is that the allocation of participants may be predictable and result in selection bias when the study groups are unmasked. However, in this trial we did not (could not) blind researchers or patients given for the intervention. Therefore, this disadvantage of block randomization was not applicable to our trial.

You have that completely wrong. The problem with permuted blocks is especially acute in unmasked trials. How would you possibly think that unmasking would somehow mitigate or eliminate this problem? Obviously, you cannot go back now and correct your mistake, but you can admit that you made a rather serious mistake, you can agree to randomize better in the future, and you can test for the extent to which your mistake resulted in selection bias. This is done with baseline p-values and also with the Berger-Exner test. See:

Berger, VW, Exner, DV (1999). “Detecting Selection Bias in Randomized Clinical Trials”, Controlled Clinical Trials 20, 319-327.

Berger, VW (2005). “Selection Bias and Covariate Imbalances in Randomized Clinical Trials”, John Wiley & Sons, Chichester, Section 6.5.

Clearly the data are not normally distributed. That could have been determined by appeal to the fact that no data are normally distributed. Why would you feel the need to test to arrive at this conclusion?

>>Our statistical analysis plan was designed to first check if data were normally distributed, so we could know which type of tests to use (parametric or non-parametric). We agree that our data are clearly not normally distributed; however, there is no harm to check this statistically. For clarity for the readers, we removed the first sentence of the section ‘statistical analysis’ for easier reading and more clarity.

There certainly is harm in allowing the folly of normality to hold sway. The data could not have possibly been normally distributed, and pretending that they could be has implications for the preservation of the Type I error rate.

Where are the p-values in Table 1?

>>Since patients were randomly distributed across groups, statistical comparison between baseline characteristics is not advocated. In short, if there would be a statistically significant difference, it would be based on chance by definition (since randomised allocation occurred). See for example the Editorial: Statistical testing for baseline differences between randomised groups is not meaningful via https://www.nature.com/articles/s41393-018-0203-y)

This is absolutely incorrect. The randomization is very much in question. Selection bias is very much a possibility. It is this selection bias, enabled by the combination of permuted blocks and unmasking, that we are testing with the baseline p-values. See:

Berger, VW (2004). “Selection Bias and Baseline Imbalances in Randomized Trials”, Drug Information Journal 38, 1-2.

Berger, VW (2005). “Selection Bias and Covariate Imbalances in Randomized Clinical Trials”, John Wiley & Sons, Chichester.

Berger, VW (2009). “Do Not Test for Baseline Imbalances Unless They Are Known To Be Present?”, Quality of Life Research 18, 399.

Berger, VW (2010). “Testing for Baseline Balance: Can We Finally Get It Right?”, Journal of Clinical Epidemiology 63, 8, 939-940.

---

## [Author Response · Author response to Decision Letter 1]

12 Mar 2021

Please see our Reply to Reviewers that was uploaded separately.

---

## [Decision Letter · Decision Letter 2]

23 Jul 2021

PONE-D-20-34585R2

Short-term Outcomes of Cochlear Implantation for Single-Sided Deafness compared to Bone Conduction Devices and Contralateral Routing of Sound hearing aids – Results of a Randomised Controlled Trial (CINGLE-trial)

PLOS ONE

Dear Dr. Peters,

Thank you for submitting your manuscript to PLOS ONE. After careful consideration, we feel that it has merit but does not fully meet PLOS ONE’s publication criteria as it currently stands. Therefore, we invite you to submit a revised version of the manuscript that addresses the points raised during the review process.

You stated, and I quote, "The randomised 526 allocation ensured equal distribution of known and unknown confounders amongst 527 groups".  This is simply not true.  It might have been true, had a better randomization method been used.  But as it stands, we cannot know this, and that is why the CONSORT guidance against baseline testing is misguided.  We do not know that all baseline imbalances are of a random nature, because they may represent selection bias.  That is what the baseline p-values would be testing.  With that, I am again asking for baseline p-values, even though CONSORY explicitly, and incorrectly, states that they are illogical.

We look forward to receiving your revised manuscript.

Kind regards,

Vance Berger

Academic Editor

PLOS ONE

Journal Requirements:

Reviewers' comments:

Reviewer's Responses to Questions

**Comments to the Author**

1. If the authors have adequately addressed your comments raised in a previous round of review and you feel that this manuscript is now acceptable for publication, you may indicate that here to bypass the “Comments to the Author” section, enter your conflict of interest statement in the “Confidential to Editor” section, and submit your "Accept" recommendation.

Reviewer #1: (No Response)

Reviewer #2: All comments have been addressed

2. Is the manuscript technically sound, and do the data support the conclusions?

Reviewer #1: (No Response)

Reviewer #2: Yes

3. Has the statistical analysis been performed appropriately and rigorously? 

Reviewer #1: (No Response)

Reviewer #2: Yes

4. Have the authors made all data underlying the findings in their manuscript fully available?

Reviewer #1: (No Response)

Reviewer #2: Yes

5. Is the manuscript presented in an intelligible fashion and written in standard English?

Reviewer #1: (No Response)

Reviewer #2: Yes

6. Review Comments to the Author

Reviewer #1: (No Response)

Reviewer #2: I have asked my statistician to comment on the concerns regarding randomization. We have had a large NIH P50 grant evaluating cochlear implant outcomes for over 30 years. Jacob Oleson has been the statistician on this grant for the last 15 years. Here is is comment on randomization in this research trial.

My opinion is that this is a reviewer who is too hung up on one particular method. It is true that the type of randomization can lead to bias. However, it is really hard (impossible) to truly randomize cochlear implants. The statistical clinical trials population (which I am not one of) spends A LOT of time discussing all of the ins and outs of the randomization process. My take is that there is always some amount of bias in studies no matter what you do. You control what can control and accept that the results are not undeniable fact. That’s what followup studies are for! A truly correctly randomized clinical trial is designed to get you close to undeniable fact. I would say accept that there could be bias (probably isn’t much) and report the p-values as requested.

The remainder of the manuscript is very well done. This is an important clinical trial and could only be done in a country that would provide hearing aids or implants to all patients. The results are dramatic and demonstrate the effectiveness of cochlear implants for single sided deafness on all variables tested. Single sided deafness is one of the most common forms of deafness that we see. The ability to treat with a cochlear implant is extremely important to this population. I suggest that this manuscript be published as soon as possible with a very high priority score.

7. PLOS authors have the option to publish the peer review history of their article (what does this mean?). If published, this will include your full peer review and any attached files.

Reviewer #1: No

Reviewer #2: No

---

## [Author Response · Author response to Decision Letter 2]

9 Aug 2021

Dear Academic Editor

In your reply dated July 23rd, you asked us to add baseline p-values. However, these values were already added to Table 1 in our most recently uploaded version dated March 12th. Based on the comments of the extra reviewer, there were no other points to improve. In the currently uploaded version, I deleted the old reply to reviewers, since no new comments needed to be addressed.

I did include the previous ‘tracked changes’-version of the manuscript, so you can clearly see the addition of the p-values in Table 1.

We hope you consider the current version of our manuscript for publication. Thanks again for your careful consideration.

Jeroen P.M. Peters, MD PhD

Corresponding author

---

## [Editor Report · Decision Letter 3]

2 Sep 2021

Short-term Outcomes of Cochlear Implantation for Single-Sided Deafness compared to Bone Conduction Devices and Contralateral Routing of Sound hearing aids – Results of a Randomised Controlled Trial (CINGLE-trial)

PONE-D-20-34585R3

Dear Dr. Peters,

We’re pleased to inform you that your manuscript has been judged scientifically suitable for publication and will be formally accepted for publication once it meets all outstanding technical requirements.

Kind regards,

Vance Berger

Academic Editor

PLOS ONE
---

## [Editor Report · Acceptance letter]

5 Oct 2021

PONE-D-20-34585R3 

Short-term Outcomes of Cochlear Implantation for Single-Sided Deafness compared to Bone Conduction Devices and Contralateral Routing of Sound hearing aids – Results of a Randomised Controlled Trial (*CINGLE-trial*) 

Dear Dr. Peters:

I'm pleased to inform you that your manuscript has been deemed suitable for publication in PLOS ONE. Congratulations! Your manuscript is now with our production department. 

Kind regards, 

on behalf of

Dr. Vance Berger 

%CORR_ED_EDITOR_ROLE%

PLOS ONE